# Rats adopt the optimal timescale for evidence integration in a dynamic environment

Alex T. Piet[1], Ahmed El Hady[1,2] & Carlos D. Brody [1,2,3]

Decision making in dynamic environments requires discounting old evidence that may no longer inform the current state of the world. Previous work found that humans discount old evidence in a dynamic environment, but do not discount at the optimal rate. Here we investigated whether rats can optimally discount evidence in a dynamic environment by adapting the timescale over which they accumulate evidence. Using discrete evidence pulses, we exactly compute the optimal inference process. We show that the optimal timescale for evidence discounting depends on both the stimulus statistics and noise in sensory processing. When both of these components are taken into account, rats accumulate and discount evidence with the optimal timescale. Finally, by changing the volatility of the environment, we demonstrate experimental control over the rats' accumulation timescale. The mechanisms supporting integration are a subject of extensive study, and experimental control over these timescales may open new avenues of investigation.

[1] Princeton Neuroscience Institute, Princeton University, Princeton 08544, USA. [2] Howard Hughes Medical Institute, Princeton University, Princeton 08544, USA. [3] Department of Molecular Biology, Princeton University, Princeton 08544, USA. Correspondence and requests for materials should be addressed to A.E.H. (email: ahady@princeton.edu) or to C.D.B. (email: brody@princeton.edu)

Decision making refers to the cognitive and neural mechanisms underlying processes that generate choices. A well characterized decision-making paradigm is that of evidence accumulation or evidence integration referring to the process by which the subject gradually processes evidence for or against different choices until making a well defined choice. Evidence accumulation is thought to underlie many different types of decisions[1] from perceptual decisions[2–4], to social decisions[5], and to value based decisions[6].

Most behavioral studies to date have focused on evidence accumulation in stationary environments. In stationary environments, the normative behavioral strategy is perfect integration[7], meaning equal weighting of all evidence across time. However, real world environments change over time. Crucially, in a dynamic environment older observations may no longer reflect the current state of the world, and an observer needs to modify their inference processes to discount older evidence. Previous studies have demonstrated that humans can modify the timescales of evidence integration, adopting leaky integration, which discounts old evidence[8,9]. These observations open many questions related to why and how subjects might alter their integration timescales, and whether the strategy adopted by humans is the optimal one. Recent studies developed a connection to normative drift-diffusion models (DDMs), and examined evidence accumulation in dynamic environments either in humans[9,10], or in ideal observer models[11]. Glaze et al.[9] found that while humans discount evidence, they did not appear to adopt the optimal discounting timescale. Here, we expand on a recently developed modeling framework[11] to demonstrate that the optimal discounting strategy depends not only on the environment's volatility, but also on the level of noise in sensory processing. Previous work[9,11] incorporated stimulus statistics into their models, but did not incorporate sensory noise. We found that once both volatility and sensory noise are taken into account, rats adopt the optimal integration timescale. We furthermore show that rats can dynamically modulate their integration timescale according to changing environmental statistics. Our findings establish rats as an adequate animal model for studying evidence accumulation and discounting in a dynamic environment.

## Results

**A dynamic decision-making task**. We developed a decision-making task that requires accumulating noisy evidence in order to infer a state that is hidden, and dynamic. Rats were trained to infer during the course of a trial, which of two states the environment was in at that moment. In each trial of our task, we first illuminate a center light inside an automated operant chamber to indicate that the rat may start the trial by nose-poking into the center port. While nose poking, auditory clicks play from speakers positioned on the left and right sides of the rat. The auditory clicks are generated from independent Poisson processes. Importantly, the left and right side Poisson rate parameters are dependent on a hidden state that changes dynamically during the course of each trial. This is in contrast to previous studies where the Poisson click rates are constant for the duration of each trial[12–14]. Within each trial, the dynamic environment is in one of two hidden states $S^1$, and $S^2$, each of which has an associated left and right click generation rate ($S^1$: rates $r_L^1$ and $r_R^1$, respectively; $S^2$: rates $r_L^2$ and $r_R^2$). In this study $S^1$ and $S^2$ were symmetric ($r_R^1 = r_R^2 =$ high rate $r_1$ and $r_R^2 = r_L^1 =$ low rate $r_2$). Each trial starts with equal probability in one of the two states, and switches stochastically between them at a fixed hazard rate $h$. On each time step, the switch probability is given by $h\Delta t$, (with $\Delta t$ kept small enough

that $h\Delta t << 1$). At the end of the stimulus period, the auditory clicks end, and the center light turns off, indicating the rat must make a left or right choice by entering one of the side reward ports. The rat is rewarded with a water drop for correctly inferring the hidden state at the end of the stimulus period (if $S^1$, go right; if $S^2$, go left). The stimulus duration is variable on each trial ($0.5-2$ s), so the rat must be prepared to infer the current hidden state at all times. Figure 1 shows a schematic of task events, as well as an example trial. Rats trained every day, performing 150–1000 self-paced trials per day.

Except where noted, the hazard rate $h = 1$ Hz, and click rates $\gamma = \log\left(\frac{r_1}{r_2}\right) = 3$, $r_1 \approx 38$ Hz, and $r_2 \approx 2$ Hz were kept constant. The click rates were chosen to be as difficult as possible while keeping rat accuracy above 70%. The chosen parameter values correspond to a high difficulty; as described below, the performance of the optimal agent for these parameters is $\approx 77\%$ (Fig. 2f). Trial difficulty is heavily modulated by the duration of time since the last hidden state change, with the hardest trials being those that end shortly after a state change. Trials had random duration with random state changes. Thus, even within one set of click rates the rats performed a broad range of trial difficulties. For analytical simplicity and consistency across the task, we therefore chose to keep click rates constant in the study.

**Optimal inference in a dynamic environment**. Here we derive the optimal, reward maximizing, procedure for inferring the hidden state. Given that each trial's duration is imposed by the experimenter and thus fixed to the rat, maximizing reward is equivalent to maximizing accuracy[7]. We build on results from ref. [11], but a basic outline is repeated here for continuity. Mathematical details can be found in Supplementary Note 3.

Before the derivation, it is worth building some intuition. Because the hidden state is dynamic, auditory clicks heard at the start of the trial are unlikely to be informative of the current state. However, because state transitions are hidden, an observer doesn't know how far back in time observations are still informative of the current state. Our derivation derives the optimal weighting of older evidence. We first consider observations in discrete timesteps of short duration $\Delta t$. Within each timestep, a momentary evidence sample $\varepsilon$ is generated. This sample is either a click on the left, a click on the right, no clicks, or a click on both sides (we will consider $\Delta t$ small enough that $r_1\Delta t << 1$ and $r_2\Delta t << 1$ so that multiple clicks are not generated within one timestep).

Following ref. [11], the probability of being in State 1 at time $t$, given all observed samples up to time $t$:

$$P(S^1|\varepsilon_{1\dots t}) \propto P(\varepsilon_t|S^1)\big((1 - h\Delta t)P(S^1|\varepsilon_{1\dots t-1}) + h\Delta t P(S^2|\varepsilon_{1\dots t-1})\big).$$
$$(1)$$

We can interpret this equation as the probability of being in State 1 given all observed evidence up to time $t$, $P(S^1|\varepsilon_{1\dots t})$, is proportional to the probability of observing the evidence sample at time $t$ given State 1, $P(\varepsilon_t|S^1)$, times the independent probability that we were in State 1 given evidence from timesteps $1\dots t-1$, $P(S^1|\varepsilon_{1\dots t-1})$. This second term has two components which depend on the probability of remaining in the same state from the last time step, $(1 - h\Delta t)P(S^1|\varepsilon_{1\dots t-1})$, and the probability of changing states after the last time step, $h\Delta t P(S^2|\varepsilon_{1\dots t-1})$.

Combining the probability of each state, we can write the posterior probability ratio ($R_t$) of the current state given all

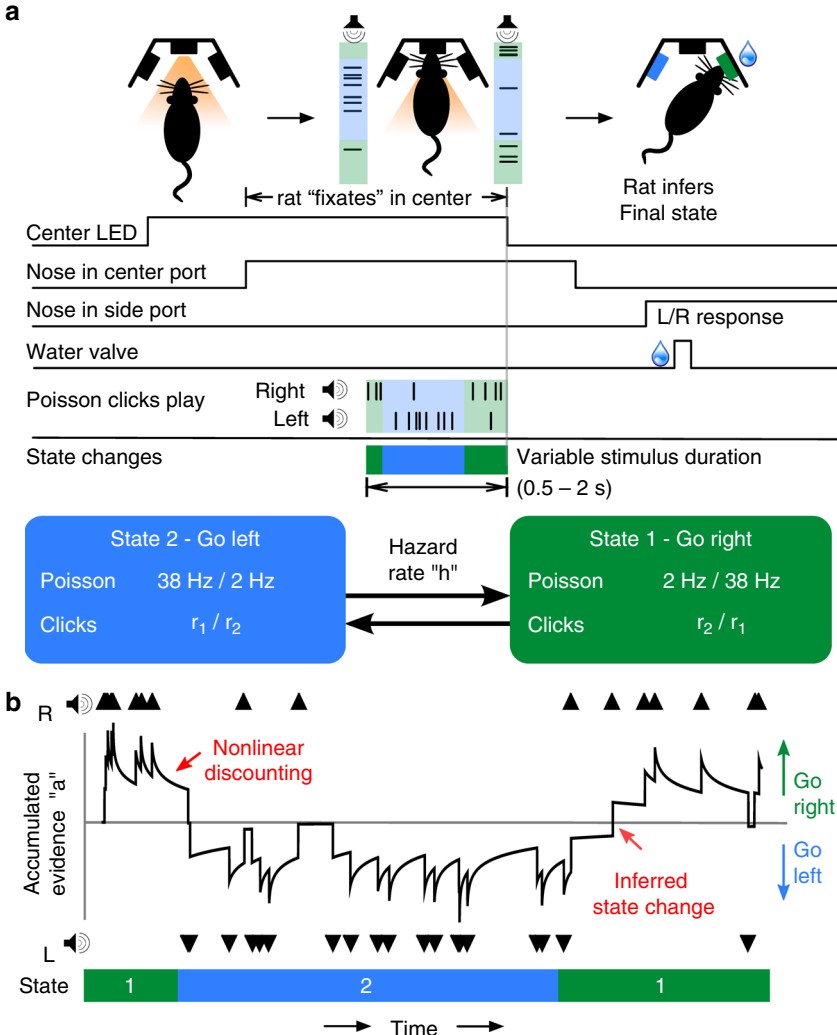

**Fig. 1** Dynamic Clicks Task structure and example trial. **a** Schematic of task events and timing. A center light illuminates indicating the rat may initiate a trial by poking its nose into a center port. Auditory clicks are generated from state-dependent Poisson processes (the two states are schematized by light green and light blue backgrounds) and played concurrently from left and right speakers. The hidden state toggles between two states according to a telegraph process with hazard rate $h$. When the auditory clicks end, and the center light turns off, the rats must infer which of the two states the trial ended in and report their decision by poking into one of two reward ports. Trials have random durations so the rat must be prepared to answer at all time points. **b** An example trial. The hidden state transitions randomly, and the auditory clicks (black triangles) are generated accordingly. The optimal inference process (black line; see text for its derivation) accumulates clicks, and discounts accumulated evidence proportionally to the volatility of the environment and click statistics. For the optimal process, a choice is generated at the end of the trial according to whether the optimal inference variable is above or below 0

previous evidence samples $\varepsilon_{1\ldots t}$:

$$R_t = \frac{P(S^1|\varepsilon_{1\ldots t})}{P(S^2|\varepsilon_{1\ldots t})} = \frac{P(\varepsilon_t|S^1)}{P(\varepsilon_t|S^2)}\left(\frac{(1-h\Delta t)R_{t-1}+h\Delta t}{(h\Delta t)R_{t-1}+1-h\Delta t}\right). \quad (2)$$

taking the limit as $\Delta t$ goes to 0:

$$d\hat{a} = \log\left(\frac{P(\varepsilon_t|S^1)}{P(\varepsilon_t|S^2)}\right) - 2h\sinh(\hat{a})dt. \quad (3)$$

Observe that in a static environment, $h = 0$, the term on the far right simplifies to $R_{t-1}$ and Eq. (2) becomes the statistical test known as the Sequential Probability Ratio Test (SPRT)[7,15,16]. When $h \neq 0$, the more complicated expression reflects the fact that previous evidence samples might no longer be informative of the current state, in a manner proportional to the environmental volatility $h$.

In order to compare Eq. (2) to standard decision-making models like the drift-diffusion model we transform the expression into a differential equation. We accomplish this by taking the logarithm of Eq. (2), then substituting $\hat{a} = \log(R)$, and finally

This differential equation describes the evolution of the log-probability ratio of being in each of the two hidden states, $\hat{a} = \log\left(\frac{P(\varepsilon_{1\ldots t}|S^1)}{P(\varepsilon_{1\ldots t}|S^2)}\right)$: $\hat{a} > 0$ indicates more evidence for $S^1$, while $\hat{a} < 0$ indicates more evidence for $S^2$. Momentary evidence samples $\varepsilon_t$ are incorporated into the log-probability ratio through the evidence term, $\log\left(\frac{P(\varepsilon_t|S^1)}{P(\varepsilon_t|S^2)}\right)$. The previously accumulated evidence is forgotten by a nonlinear discounting term, $-2h\sinh(\hat{a})$ (See Fig. 2c). The evidence discounting reduces the effect of older evidence, weighting recent evidence more. This discounting reflects the fact that older evidence may no longer be

informative of the current state of the environment. In a static environment, $h = 0$, the discounting term is eliminated, and the ideal observer perfectly integrates the momentary evidence samples. In analysis of the static decision-making models, the evidence term is commonly approximated by its expectation (drift) and variance (diffusion), transforming Eq. (3) into the Drift-Diffusion Model (DDM) for decision making[7]. In order to develop a deeper understanding of the optimal inference on our task, we precisely evaluate the evidence term for each discrete Poisson evidence sample. This exact evaluation is not possible in previous decision-making tasks. In a small sample window of duration $\Delta t$, the probability of a Poisson event is $r\Delta t$, where $r$ is the parameter of the Poisson process (provided $r\Delta t \ll 1$). In our task a momentary sample $\varepsilon_t$ is the result of two independent Poisson processes and can take on four possible values: a click on both sides, a click on the right, a click on the left, or no clicks. Evaluating the evidence term for these four conditions:

A click on both sides

$$\log\frac{P(\varepsilon_t|S^1)}{P(\varepsilon_t|S^2)} = \log\frac{P(\text{click}-\text{R}|S^1)P(\text{click}-\text{L}|S^1)}{P(\text{click}-\text{R}|S^2)P(\text{click}-\text{L}|S^2)}$$
$$= \log\frac{(r_1\Delta t)(r_2\Delta t)}{(r_2\Delta t)(r_1\Delta t)} = 0. \tag{4}$$

No clicks

$$\log\frac{P(\varepsilon_t|S^1)}{P(\varepsilon_t|S^2)} = \log\frac{P(\text{no}-\text{click}-\text{R}|S^1)P(\text{no}-\text{click}-\text{L}|S^1)}{P(\text{no}-\text{click}-\text{R}|S^2)P(\text{no}-\text{click}-\text{L}|S^2)}$$
$$= \log\frac{(1-r_1\Delta t)(1-r_2\Delta t)}{(1-r_2\Delta t)(1-r_1\Delta t)} = 0. \tag{5}$$

A click on the right

$$\log\frac{P(\varepsilon_t|S^1)}{P(\varepsilon_t|S^2)} = \log\frac{P(\text{click}-\text{R}|S^1)P(\text{no}-\text{click}-\text{L}|S^1)}{P(\text{click}-\text{R}|S^2)P(\text{no}-\text{click}-\text{L}|S^2)}$$
$$= \log\frac{(r_1\Delta t)(1-r_2\Delta t)}{(r_2\Delta t)(1-r_1\Delta t)} \equiv +\kappa(r_1, r_2). \tag{6}$$

A click on the left

$$\log\frac{P(\varepsilon_t|S^1)}{P(\varepsilon_t|S^2)} = \log\frac{P(\text{no}-\text{click}-\text{R}|S^1)P(\text{click}-\text{L}|S^1)}{P(\text{no}-\text{click}-\text{R}|S^2)P(\text{click}-\text{L}|S^2)}$$
$$= \log\frac{(1-r_1\Delta t)(r_2\Delta t)}{(1-r_2\Delta t)(r_1\Delta t)} \equiv -\kappa(r_1, r_2). \tag{7}$$

We define the function $\kappa(r_1, r_2)$ to be the increase in the log-probability ratio from the arrival of a single click on the right, given click rates $r_1$, $r_2$. The function $\kappa$ tells us how reliably each click indicates the hidden state. This is easily seen when letting $\Delta t \to 0$, so $\kappa \to \log\frac{r_1}{r_2}$. If the click rates $r_1$ and $r_2$ are very similar (so $\kappa$ is small) then we expect many distractor clicks (clicks from the smaller click rate that do not indicate the correct state), so an individual click tells us little about the underlying state. On the other hand, if the click rates are very different (so $\kappa$ is large) then we expect very few distractor clicks, so an individual click very reliably informs the current state. In the limit of one of the click rates going to zero: $\kappa \to \infty$, and a single click tells us the current state with absolute certainty. In our task, the two click rates $r_1$ and $r_2$ always sum to 40 Hz. Figure 2a shows $\kappa$ as a function of the click rates.

Re-writing the log-evidence term in Eq. (3) in terms of $\kappa$ and using $\delta_{L/R,t}$ to represent the left/right click times, we can summarize across all four conditions:

$$d\hat{a} = \kappa(r_1, r_2)\left(\delta_{R,t} - \delta_{L,t}\right) - 2h\sinh(\hat{a})dt. \tag{8}$$

We can then rescale Eq. (8) by $\kappa$, let $a = \frac{\hat{a}}{\kappa} = \frac{\log(R)}{\kappa}$, to put our evidence accumulation equation in units of clicks:

$$da = \delta_{R,t} - \delta_{L,t} - \frac{2h}{\kappa}\sinh(\kappa a)dt. \tag{9}$$

Equation (9) has a simple interpretation, sensory clicks are integrated, $\delta_{R,t} - \delta_{L,t}$, while accumulated evidence is discounted, $-\frac{2h}{\kappa}\sinh(\kappa a)$, proportionally to the volatility of the environment, $h$, and the reliability of each click, $\kappa$. This interpretation also allows for a simple assay of behavior: do rats adopt the optimal discounting timescale? We will present two quantitative methods for measuring the rats discounting timescales. However, before examining rat behavior, we need to examine the impact of sensory noise on optimal behavior.

**Sensory noise decreases click reliability**. The function $\kappa(r_1, r_2)$ tells us how reliably each click indicates the underlying state as a function of the click generation rates $r_1$ and $r_2$. The computation above of $\kappa$ assumes that each click is detected and correctly localized as either a left or right click with perfect accuracy. Previous studies using pulse-based evidence demonstrate that rats have significant sensory noise[12,17]. Sensory noise was measured through parametric models that included a parameter noise due to each pulse of evidence. The exact biological origin of this noise remains unclear. Regardless of its origins, sensory noise is a significant component of rodent behavior. However, previous inference models in dynamic environments have not incorporated sensory noise.[9,11]

We now show that sensory noise decreases the reliably of each click. While sensory noise can be modeled in many ways, primarily the mislocalization of clicks changes the click reliability. We analyze the cases of Gaussian noise on the click amplitudes and missing clicks, and provide a general argument for mislocalization in Supplementary Note 7. Mislocalization refers to how often clicks are incorrectly localized to the other speaker (hearing a click from the left and assigning it to the right). For intuition, consider that if a rat could never tell whether a click was played from the right or left then each click would never indicate any information about the underlying state. We again evaluate the log-evidence term, now including the probability of click mislocalization, $n$:

A click on the right

$$\log\frac{(r_1\Delta t)(1-n)(1-r_2\Delta t) + (1-r_1\Delta t)(r_2\Delta t)(n)}{(r_2\Delta t)(1-n)(1-r_1\Delta t) + (1-r_2\Delta t)(r_1\Delta t)(n)} = +\kappa(r_1, r_2, n). \tag{10}$$

A click on the left

$$\log\frac{(r_2\Delta t)(1-n)(1-r_1\Delta t) + (1-r_2\Delta t)(r_1\Delta t)(n)}{(r_1\Delta t)(1-n)(1-r_2\Delta t) + (1-r_1\Delta t)(r_2\Delta t)(n)} = -\kappa(r_1, r_2, n). \tag{11}$$

The terms for no clicks, or clicks on both sides evaluate to 0. As in the case with no sensory noise, the log-evidence is either 0, or has value $\kappa$. We can simplify the expression for $\kappa$ by letting $\Delta t \to 0$:

$$\kappa(r_1, r_2, n) = \log\frac{r_1(1-n) + r_2 n}{r_2(1-n) + r_1 n}. \tag{12}$$

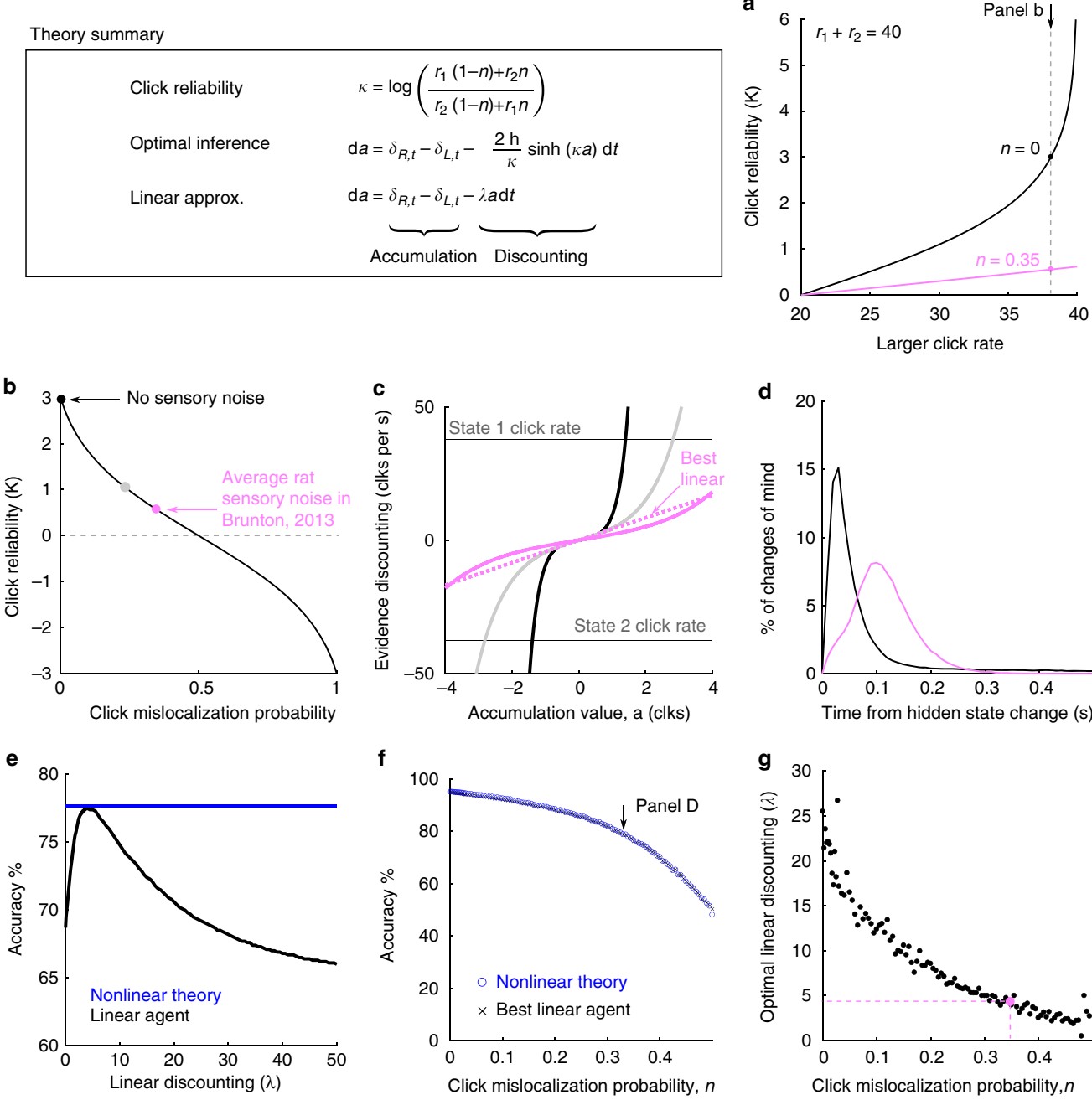

**Fig. 2** Optimal discounting rates depends on click reliability and can be well-approximated by linear discounting. **a, b** The reliability $\kappa$ of each click depends on the Poisson click rates $r_1$ and $r_2$, as well as the click mislocalization probability $n$. **a** $\kappa$ as a function of click rates with no sensory noise (black), and with the average level of sensory noise reported in Brunton, 2013[12] (pink). Vertical dashed line shows the click rates used in the study. **b** The reliability of each click depends on how consistently each click can be correctly localized to the side that generated it. At 50% mislocalization each click contains no information about the current state, so $\kappa = 0$. The pink dot uses the average level of sensory noise. The grey dot uses an intermediate level of the sensory noise. All values displayed here use the click rates used in the study (38 Hz/2 Hz). **c** Discounting functions for the three sensory noise levels in (**b**) (same colors). Increasing sensory noise causes the discounting functions to weaken. Horizontal lines show average clks/sec in each of the two states.
**d** Histogram of changes of mind produced by the optimal inference equation. Timing is relative to the last change in the hidden state. (Black) Inference without sensory noise, (pink) inference with average rat level of sensory noise. **e** The optimal nonlinear discounting function can be approximated by a linear discounting function. If the linear discounting function is tuned appropriately, accuracy is close to the full nonlinear function. **f** Comparison between optimal nonlinear discounting function (blue) and the best linear approximation (black), in terms of average accuracy for different noise levels. The best linear approximation is effectively equivalent. Arrow indicates parameter values used in (**d**). **g** The best linear discounting rate $\lambda$ as a function of sensory noise. Increasing sensory noise decreases the discounting rate. The best linear function is found numerically on a set of 30k trials, which produces some variability for different noise levels. Pink dot indicates average rat sensory noise

Sensory noise decreases how reliably each click informs the underlying state in the trial, increasing $n$ decreases $\kappa$. If $n = 0$, we recover the original $\kappa$ derived without noise. If $n = 0.5$, then each click is essentially heard on a random side, and therefore contains no information so $\kappa = 0$. If $n = 1$, then we simply flip the sign of all clicks.

Previous studies using the same auditory clicks have shown that rats have significant sensory noise[12,17]. We computed an estimate of the average sensory noise for the rats in ref. [12], finding an average value of $n = 0.35$. Figure 2a, b shows $\kappa$ against $n$ and click rates $r_1$, $r_2$, and highlights the average rat sensory noise from ref. [12]. Click mislocalization for each rat was estimated by fitting a parametric model introduced below and in ref. [12]. See methods for estimation of click mislocalization from model parameters. Two lines of evidence suggest that our estimate of the level of sensory noise is reliable. First, ref. [12] predicted performance on single-click trials based on each rat's sensory noise (Fig. 3b, ref. [12]). Second, we found click mislocalization levels were constant across a wide range of click rates in ref. [12] (Supplementary Note 6). Figure 2a highlights that sensory noise is the dominant factor on determining click reliability.

**Lower click reliability requires integrating longer**. The discounting term of Eq. (9) has $\kappa$ in the denominator as well as the argument of the sinh term. As a result, it is not clear how decreasing the click reliability $\kappa$ changes the behavior of the optimal inference agent. To gain insight, consider that if evidence is very reliable, accurate decisions can be made by only using a few clicks from a small time window. However, if evidence is unreliable, a longer time window must be used to average out unreliable clicks. This intuition is confirmed by plotting the discounting function for a variety of evidence reliability values (Fig. 2c). Decreasing reliability weakens the evidence discounting term creating longer integration timescales.

**Evidence discounting leads to changes of mind**. The optimal inference equation attempts to predict the hidden state. As the hidden state dynamically transitions, we expect the inference process to track, albeit imperfectly, the dynamic transitions. From the perspective of a subject this dynamic tracking leads to changes of mind in the upcoming choice. Through the optimal inference process we can predict the timing of changes of mind by looking for times when the sign of the inference process changes (sign($a$)). We simulated the optimal inference agent on a large dataset of trials assuming either no sensory noise (black), or average rat sensory noise (pink). For both agents we computed the distribution of when changes of mind happen relative to changes in the hidden state of the trial. Figure 2d shows the predicted timing of changes of mind with and without sensory noise, and the temporal relationship to hidden state changes. As expected, hidden state changes trigger changes of mind with a temporal delay that increases with sensory noise.

**Linear approximation to discounting function is accurate**. The full nonlinear discounting function, $-\frac{2h}{\kappa}\sinh(\kappa a)$, is complicated and difficult to interpret. To aid our analysis of rat behavior, we focus on the accumulation timescale and consider a linear approximation to the discounting function, $-\lambda a$, where $\lambda$ gives the discounting rate. There are many possible linear approximations with different slopes. A linear approximation using the slope of sinh at the origin will fail to capture the strong discounting farther from the origin. The best approximation is the one that achieves the highest accuracy at predicting the underlying state. We found the best linear approximation numerically.

Figure 2e shows, for a particular noise level and click rates, the accuracy of a range of linear discounting agents against the full nonlinear agent. If $\lambda$ is tuned correctly, the linear agent accuracy is very close to the full nonlinear function. We find this to be true across a wide range of noise values (Fig. 2f). While the optimal linear discounting strength at each noise level changes (Fig. 2g), the accuracy is always very close to that of the full nonlinear theory. For the average level of sensory noise, we find the linear agent to have 99.8% of the accuracy of the nonlinear model. The linear model was optimized on a training set of trials, and both models were evaluated on a test set of held-out trials and achieved 77.15 and 77.34% accuracy respectively. When given identical sensory noise, the trial-by-trial choices between the linear and nonlinear models agree on 97% of trials. It is important to note that a linear approximation in general will not always be close in accuracy to the full nonlinear theory[11], but for our specific click rate parameters it is an accurate approximation.

Given that a linear discounting function matches the accuracy of the nonlinear model, we analyze rat evidence discounting behavior by looking for the appropriate discounting rate or equivalently the appropriate integration timescale. Specifically, we compare the rat behavior to this linear discounting equation:

$$\mathrm{d}a = \delta_{\mathrm{R},t} - \delta_{\mathrm{L},t} - \lambda a \,\mathrm{d}t, \qquad (13)$$

where $\lambda$ is the discounting rate and $\frac{1}{\lambda}$ is the integration timescale. We did not examine whether rats demonstrate nonlinear evidence discounting because the linear approximation matches the accuracy of the nonlinear theory.

**Reverse correlation reveals the integration timescale**. Psychophysical reverse correlation is a statistical method to find what aspects of a behavioral stimulus influence a subject's choice. Here we use reverse correlation to find the integration timescale used by the rats. We normalized the reverse correlation curve to have an area under the curve equal to one. This step lets the curves be interpreted in units of effective weight at each time point. A flat reverse correlation curve indicates even weighting of evidence across all time points. Previous studies in a static environment find rats with flat reverse correlation curves[12–14]. Figure 3a shows the reverse correlation for an example rat in a dynamic environment. The stimulus earlier in the trial is weighted less than the stimulus at the end of the trial indicating evidence discounting. Figure 3b shows the mean reverse correlations for all rats in the study. Figure 3c shows the reverse correlation curves from a family of linear discounting agents, $\mathrm{d}a = \delta_{\mathrm{R}} - \delta_{\mathrm{L}} - \lambda a \,\mathrm{d}t$, with $\lambda$ ranging from 0 to 30. The curves were generated from a synthetic dataset of 20,000 trials. The weaker the discounting rate, the flatter the reverse correlation curves. To quantify the discounting timescale from the reverse correlation curves, an exponential function $e^{bt}$ was fit to each curve. The parameter $b$ reliably recovers the discounting rate $\lambda$ (Fig. 3d).

**Rats adapt to the optimal timescale**. To compare each rat's evidence discounting timescale to the optimal inference equation, we simulated the optimal linear inference agent on the trials each rat experienced. We then computed the reverse correlation curves for both the rats and the optimal linear agent (Fig. 4a). We then fit an exponential function to each of the reverse correlation curves. Rat behavior was compared with two optimal agents. The first optimal linear agent assumes no sensory noise; while the second agent uses the optimal timescale given the average level of sensory noise across rats reported in ref. [12] (Fig. 4b). When the average level of sensory noise is taken into account, the rats match the optimal timescale. The reverse correlation analysis

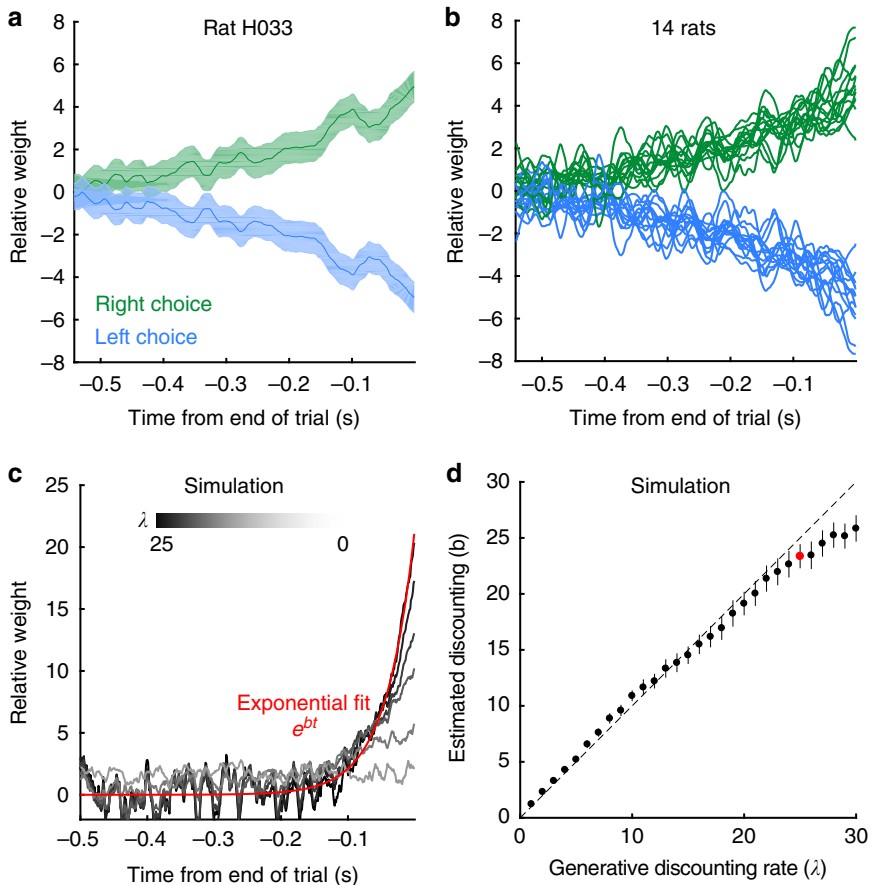

Fig. 3 Rats discount evidence. **a** Reverse correlation curves for an example rat reveals how clicks at each time point influence the rat's decision. Error bars show standard deviation. **b** Reverse correlation curves for 14 rats. Error bars are omitted for clarity. **c** Reverse correlation curves for a range of simulated linear discounting agents. Black to white lines indicate increasing discounting rates ($\lambda$). Only the reverse correlation curve for the right choice are shown for clarity. Each curve was fit with an exponential function (example red). The fit parameters are used in (**d**). **d** Exponential fit to each discounting agent recovers the generative linear discounting rate. Example in (**c**) show with red dot. Error bars show 95% confidence intervals of exponential fit

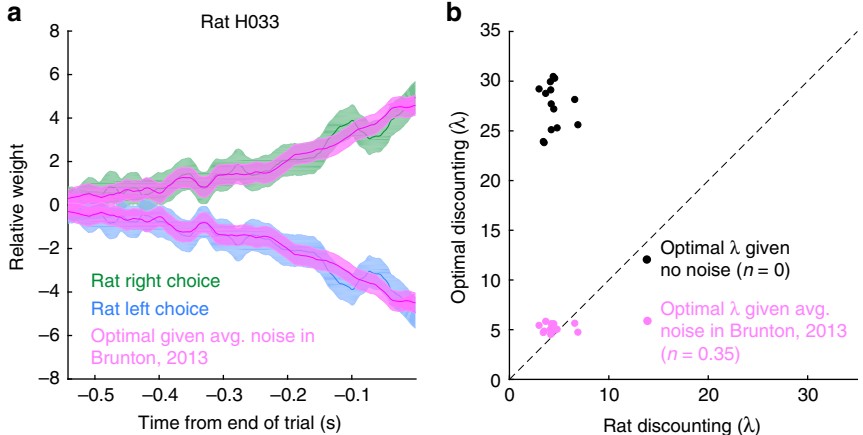

Fig. 4 Rats discount evidence with the optimal timescale. **a** Example reverse correlation curve for one rat (blue and green), and the reverse correlation curve from the optimal linear inference agent (pink) with the average rat sensory noise from Brunton, 2013[12]. The optimal linear inference agent was simulated on the same trials the rat performed. Shaded area shows one standard deviation. **b** Quantification of discounting timescales. Rat integration timescales plotted with optimal agents with no sensory noise (black), or with sensory noise (pink). The variability in optimal discounting rates is a result of measuring the reverse correlation curves on a different set of trials each rat actually performed. Error bars omitted for clarity

shows that rats are close to the optimal timescale given the average level of sensory noise in a separate cohort of rats.

**A trial-by-trial behavioral model captures rat behavior**. In order to examine individual variations in noise level and integration timescales, we fit a behavioral accumulation of evidence model from the literature to each rat[12–14]. This model generates a moment-by-moment estimate of a latent accumulation variable. The dynamical equations for the model are given by:

$$da = \left( \delta_{R,t} \cdot \eta_R \cdot C - \delta_{L,t} \cdot \eta_L \cdot C \right) dt - \lambda a dt + \sigma_a dW, \quad (14)$$

$$\frac{dC}{dt} = \frac{1 - C}{\tau_\phi} + (\phi - 1)C\left( \delta_{R,t} + \delta_{L,t} \right). \quad (15)$$

At each moment in a trial, the model generates a distribution of possible accumulation values $P(a|t, \delta_R, \delta_L)$. In addition to the click integration and linear discounting that was present in our normative theory, this model also parameterizes many possible sources of noise. Each click has multiplicative Gaussian sensory noise, $\eta_{L/R} = \mathcal{N}\left(1, \sigma_s^2\right)$. In addition to the sensory noise, each click is also filtered through an adaptation process, $C$. The adaptation process is parameterized by the adaptation strength $\phi$, and a adaptation time constant $\tau_\phi$. If $\phi > 1$ the model has facilitation of sequential clicks, and if $\phi < 1$ the model has depression of sequential clicks. The accumulation variable $a$ also undergoes constant additive Gaussian noise $W$ with variance $\sigma_a^2$. Finally, the initial distribution of $a$ has some initial variance given by $\sigma_i$. See ref. [12] for details on the development and evaluation of this model. The only modification to the model from previous studies is the removal of the sticky bounds $B$, which are especially detrimental to subject performance given the dynamic nature of the task. This model is a powerful tool for the description of behavior on this task because of its flexibility at characterizing many different behavioral strategies[12–14]. We parameterized the model with linear discounting, rather than nonlinear discounting in the full theory for three reasons. First, the linear discounting model has been fit to rat behavior in static environments, allowing a direct comparison to previous results. Second, the linear model has an analytical solution that greatly facilitates analysis. Third, the linear model has comparable accuracy to the nonlinear model with less parameters, simplifying the fitting procedure and providing a more parsimonious description of rat behavior.

The model was fit to individual rats by maximizing the likelihood of observing the rat's choice on each trial. To evaluate the model, we compared the reverse correlation curves from the model and subject. Figure 5a shows the comparison for an example rat, showing that the model captures the timescale of evidence discounting seen by the reverse correlation analysis. To evaluate parameter sensitivity in our model, we approximated the local likelihood landscape by the Hessian matrix. The inverse of the Hessian matrix was then used as an estimate of the parameter covariance[18]. Supplementary Table 1 shows parameter values and parameter uncertainty for each rat. We used the eigenvalue decomposition of each rat's Hessian matrix to assess whether parameters in our model trade off against each other. Eigenvectors significantly aligned with multiple parameters can indicate trade-offs in the likelihood landscape. We found no significant trade-offs involving the discounting parameter $\lambda$ (Supplementary Figure 22). Finally, we plotted the residual error plots for each rat to identify systematic errors by the model. For each rat, the residual error was constant in time, indicating our model fit short

and long duration trials equally well (Supplementary Figures 20, 21).

In order to analyze the model fits we compared the best fit parameters for each rat with those from rats trained on the static version of the task (data from ref. [12]). The evidence discounting strength parameter $\lambda$ shows a striking difference between the two rat populations (Fig. 5b). In the static task, the rats have small discounting rates indicating an integration timescale comparable to the longest trial the rats experienced[12–14]. In the dynamic task, the rats have strong evidence discounting, consistent with the reverse correlation analysis.

To assess whether rats individually calibrate their discounting timescales to their level of sensory noise, we estimated the sensory noise level from the model parameters. Figure 5b shows each rat's fit compared to the numerically obtained optimal linear discounting levels from Fig. 2b. The rats appear to have slightly larger discounting rates than predicted by the normative theory. The deviation from the normative theory may be due to other parameters in the behavioral model, the fact that we considered only the average level of sensory adaptation, or other factors. In order to more directly examine whether the rats were adopting the optimal timescale, we asked whether the rat's discounting rates were constrained by the other model parameters. For each rat, we took the best fitting model parameters, and froze all parameters except the discounting rate parameter $\lambda$. Then, we found the value of $\lambda$ that maximized accuracy on the trials each rat performed. Note this optimization did not ask to maximize the similarity to the rat's behavior. We found that given the other model parameters, the accuracy maximizing discounting level was very close to the rat's discounting level (Fig. 5d) meaning that other model parameters highly constrain the rats' discounting rates. Further, while the discounting rates changed slightly, the improvement in total trial accuracy changed even less. For all rats, optimizing the discounting rate increased the total accuracy of the model by less than 1% (Fig. 5e). Taken together these results suggest that rats discount evidence at the optimal timescale.

**Individual rats in different environments**. Previous studies have demonstrated that rats can optimally integrate evidence in a static environment[12]. Here we demonstrated that rats use the optimal timescale for evidence integration and discounting in a dynamic environment. But can individual rats change their integration timescale, to match the volatility of different environments? To probe this question, we moved four rats from a dynamic environment ($h = 0.5$ Hz) to a static environment ($h = 0$ Hz), and then back. The rats trained in each environment for many daily sessions (minimum 25 sessions). In each environment, we quantified the rats' behavior using the model of Eqs. (14) and (15) above. Figure 6b shows the recovered evidence discounting parameter from a model fit to a dataset combined across all rats. For this fit, we split each rat's trials from each environment into a first and a second half in time, and used only the second half, so as to assess behavior subsequent to any transient effects due to the transition to a new environment. Consistent with our normative theory, rats in the $h = 0.5$ Hz environment show discounting rates approximately half the strength of rats in the $h = 1$ Hz environment. One of the rats provided a large number of trials per session ($n = 58{,}427$ 0 Hz trials), which gave us the opportunity to examine how quickly it adapted to the new environment, by fitting the trial-by-trial model in consecutive blocks of 7500 trials. Only the evidence discounting parameter $\lambda$ was fit to each block, the other parameters we fit separately to the first set of 0.5 Hz trials. Figure 6 shows the evidence discounting parameter on each of these 7500 blocks. Evidence discounting is stable before the switch, and then begins to adjust on the first block of trials, reaching the new

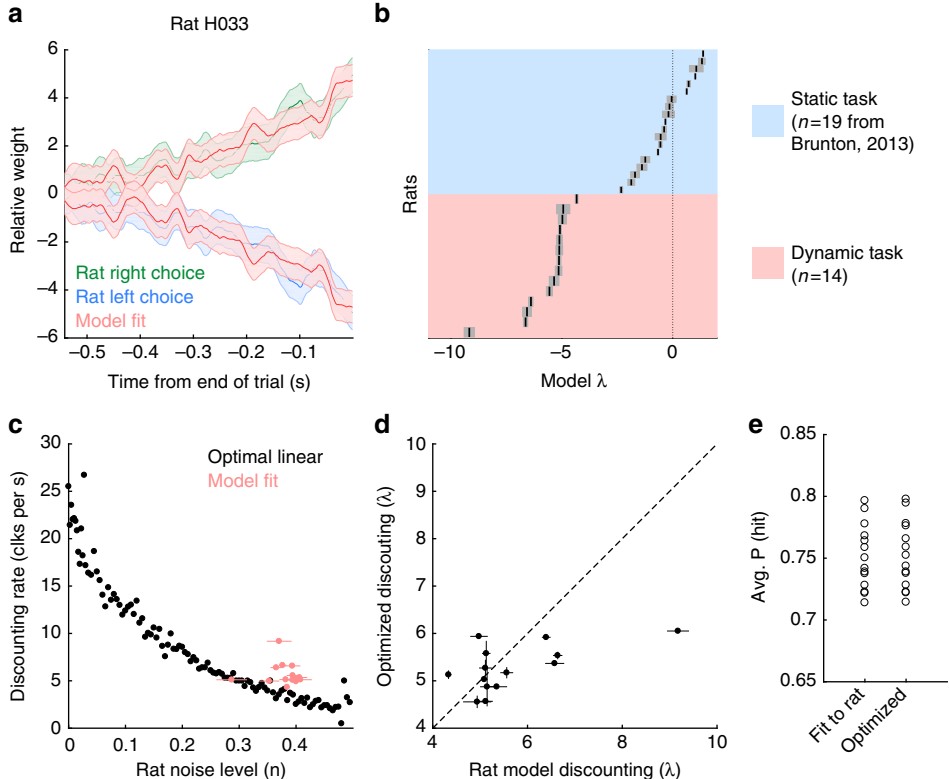

**Fig. 5** Trial-by-trial model captures rat behavior, and shows optimal linear discounting. **a** Example reverse correlation curves generated by the trial-by-trial model (light red) compared with a rat's behavior (blue and green). Shaded area is one standard deviation. **b**–**d** Error bars show 95% confidence intervals. **b** Best fitting discounting rates for rats trained on the dynamic task (light red), and for rats trained in a static environment (blue, data and fits from Brunton, 2013[12]). **c** Each rat's noise level and discounting rate (light red) compared to the optimal linear trade-off (black). **d** Each rat's evidence discounting parameter compared to the accuracy maximizing discounting level. **e** The average accuracy for the model fit to each rat's behavior, and optimized to maximize accuracy

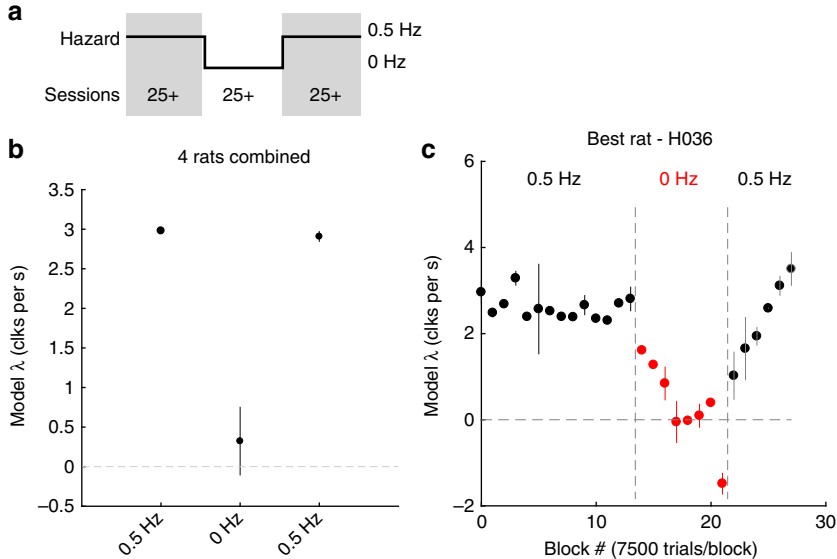

**Fig. 6** Rats adapt to changing environmental conditions. **a** Schematic outlining the experimental design. Four rats were moved from a 0.5 Hz hazard rate to 0 Hz, then back to 0.5 Hz. Rats stayed in each environment for multiple daily training sessions, with a minimum of 25 sessions. **b** Quantification of evidence discounting rates before, during, and after the switch for the combined rat dataset using the trial-by-trial model. **c** Evidence discounting rate during the switch for one rat using the trial-by-trial model on consecutive blocks of 7500 trials. Only the evidence discounting term was fit. **b**, **c** Error bars show 95% confidence intervals

optimal value on the 4th block of 7500 trials. In sum, the data from this experiment indicate that rats can gradually adjust their temporal integration behavior to match different environments.

## Discussion

Previously accumulation of evidence has been studied in a stationary environment. These studies have given behavioral and neural insights into the ability of rats, monkeys, and humans to optimally accumulate evidence[12,19–24]. These studies have showed that subjects can gradually accumulate evidence for decision-making, adopting long integration timescales. Trial-by-trial modeling[12] revealed that errors originated from sensory noise, not from the evidence accumulation process. Using a high-throughput automated rat training, we trained rats to accumulate and discount pulse-based evidence in a dynamic environment. Extending results from the literature[9,11], we formalized the optimal behavior on our task, which critically involves discounting evidence on a timescale proportional to the environmental volatility and the reliability of each click. Importantly, our pulse-based analysis allowed us to separate evidence reliability into experimenter imposed stimulus statistics and sensory noise. We find that once sensory noise is taken into account, the rats have timescales consistent with the optimal inference process. We used behavioral modeling to investigate rat to rat variability, and to predict a moment-by-moment estimate of the rats' accumulated evidence. Finally, we demonstrated rats can adjust their integration timescales in response to changing environmental statistics. Our findings open new questions into complex rodent behavior and the underlying neural mechanisms of decision making.

Glaze et al.[9] examined human decision making in a dynamic environment, and found that humans show nonlinear evidence discounting, but do not match the optimal inference process. Quantifying the subject's estimates of environmental volatility, rather than the discounting rate, they found that subjects typically underestimated the volatility (Fig. 7 of ref. [9]). This finding is consistent with the role of sensory noise decreasing the discounting rate. Incorporating models of human sensory noise into their analysis could potentially explain deviations from optimality. However, our rats performed more trials than the human subjects in ref. [9]. Human subjects with more experience may more closely match optimal processes. Unlike ref. [9], we did not examine whether our subjects demonstrated nonlinear evidence discounting because the linear approximation in our task is very close to the nonlinear theory (Fig. [2]). Our linear approximation does not parameterize environmental volatility, so we did not estimate our rat's estimate of this parameter. However, with the estimate of sensory noise from ref. [12] we could accurately predict the rat's integration timescales without considering volatility. Supplementary Figure 18 directly examines whether poor estimates of volatility could explain rat behavior. We find that in a high click reliability environment, where volatility estimates should not influence integration timescales, rats discount consistent with the sensory noise predicted timescale.

Our study has several potential limitations. First, the presence of sensory noise complicates the analysis of behavioral and neural data. Experimental modulation of sensory noise would provide a more direct test of our findings. While such a manipulation would be insightful, it falls outside the scope of this study. Lower sensory noise would facilitate investigations with harder click rates, and potentially distinguish the nonlinear and linear discounting models. Second, this study tested a limited range of click rates and environmental volatilities. Rodent behavior may deviate from the optimal timescale in different parameter regimes where other factors influence behavior. Future work should investigate

how integration timescales change over a broader parameter regime. Third, because the linear and nonlinear inference processes are very similar in our parameter regime we focused on the measurement of the integration timescale. Future work should utilize a broader parameter regime or lower sensory noise to investigate whether rodent behavior matches the nonlinear theory.

Rodent models facilitate the use of a wide range of experimental tools to investigate the neural mechanisms underlying behavior. Our task will facilitate the investigation of two neural mechanisms. First, due to the dynamic nature of each trial, subjects change their mind often during each trial allowing experimental measurement of changes of mind driven by internal estimates of accumulated evidence. Previous studies of rat decision making have identified a cortical structure, the frontal orienting fields (FOF) as a potential substrate for upcoming choice memory[13,14,25–27]. Future work could investigate if and how the FOF tracks upcoming choice in a dynamic environment during changes of mind. It will complement already neurophysiological studies of changes of mind[28,29].

Second, normative behavior in a dynamic environment requires tuning the timescale of evidence integration to the environmental volatility. There is a large body of experimental and theoretical studies on neural integrator circuits[30–33]. Many circuit mechanisms have been proposed, from random unstructured networks[34,35], feed-forward syn-fire chains[31], and recurrent structured networks of many forms[30,36,37]. The task developed here allows for experimental control of the putative neural integrator's timescale within the same subject. Measurement of neural activity in different dynamic environments, and thus different integration timescales, may shed light into which mechanisms are used in neural circuits for evidence integration. For instance, unstructured networks, or feed-forward networks may re-tune themselves via adjusting read-out weights. Networks that integrate via recurrent dynamics; however, would re-tune themselves via changes in those recurrent dynamics. Alternatively, measurement of neural activity in different dynamic environments may reveal fundamentally new mechanisms of evidence integration. For instance, ref. [13] proposed multiple integration networks with different timescales to account for behavioral changes in response to cortical inactivations. Our task may allow further investigation into the structure and dynamics of neural integrators.

## Methods

**Subjects.** Animal use procedures were approved by the Princeton University Institutional Animal Care and Use Committee and carried out in accordance with NIH standards. All subjects were adult male Long Evans rats (Vendor: Taconic and Harlan, USA) placed on a controlled water schedule to motivate them to work for a water reward.

**Behavioral training.** We trained 14 rats on the dynamic clicks task (Fig. [1]). Rats went through several stages of an automated training protocol. In the final stage, each trial began with an LED turning on in the center nose port indicating to the rats to poke there to initiate a trial. Rats were required to keep their nose in the center port (nose fixation) until the light turned off as a go signal. The center-fixation period lasted 2 s on all trials. During center fixation, auditory cues were played indicating the current hidden state. The duration of the stimulus period ranged from 0.5 to 2 s. After the go signal, rats were rewarded for entering the side port corresponding to the hidden state at the end of the stimulus period. The hidden state did not change after the go signal. A correct choice was rewarded with 24 μl of water; while an incorrect choice resulted in a punishment noise (spectral noise of 1 kHz for a 0.7 s duration). The rats were put on a controlled water schedule where they receive at least 3% of their weight every day. Rats trained each day in a training session on average 120 min in duration. Training sessions were included for analysis if the overall accuracy rate exceeded 70%, the center-fixation violation rate was below 25%, and the rat performed more than 50 trials. In order to prevent the rats from developing biases towards particular side ports an anti-biasing algorithm detected biases and probabilistically generated trials with the correct answer on the non-favored side.

**Linear discounting agents**. To analyze the performance of linear discounting agents at varying levels of noise, we created synthetic noisy-datasets. For each level of click noise, each click switched sides according to the noise level. On each of these datasets, we numerically optimized the discounting level that maximized the accuracy of predicting the hidden state at the end of the trial.

**Psychophysical reverse correlation**. The computation of the reverse correlation curves was very similar to methods previously reported[12–14]. However, one additional step is included to deal with the hidden state. The first step is to smooth the click trains on each trial with a causal Gaussian filter, $k(t)$, this creates one smooth click rate for each trial. The filter had a standard deviation of 5 ms.

$$r_i(t) = \delta_{R,t} * k(t) - \delta_{L,t} * k(t) \tag{16}$$

Then, the smooth click rate on each trial was normalized by the expected click rate for that time step, given the current state of the environment. This gives us the deviation (the excess click rate) from the expected click rate for each trial.

$$e_i(t) = r_i(t) - \langle r(t)|S_i(t)\rangle \tag{17}$$

Finally, we compute the choice triggered average of the excess click rate by averaging over trials based on the rat's choice.

$$\text{excess rate}(t|\text{choice}) = \langle e(t)|\text{choice}\rangle \tag{18}$$

The excess rate curves were then normalized to integrate to one. This was done to remove distorting effects of a lapse rate, as well to make the curves more interpretable by putting the units into effective weight of each click on choice. To quantify the timescale of the reverse correlation curves, we fit an exponential of the form $ae^{bt}$ to each curve. The parameter $a$ is a scale parameter. The parameter $b$ is the discounting rate, while $1/b$ is the integration timescale. All exponential fits were computed using the MATLAB package *fit*, which used a least squares fit on a linear scale. 95% Confidence intervals on the exponential fits are shown in Supplementary Figure 15 and calculated by the *fit* package.

**Behavioral model**. Previous studies using this behavioral accumulation of evidence model[12] have included sticky bounds which absorb probability mass when the accumulated evidence reaches a certain threshold. We found this sticky bounds to be detrimental to high performance on our task, so we removed them. The removal of the sticky bounds facilitates an analytical solution of the model. The model assumes an initial distribution of accumulation values $P(a|t = 0) = \mathcal{N}(\mu_0, \sigma_i^2)$. At each moment in the trial, the distribution of accumulation values $P(a|t, \delta_R, \delta_L)$ is Gaussian distributed with mean $\mu$ and variance $\sigma^2$ given by:

$$\mu(t) = \mu_0 e^{\lambda t} + \int_0^t \left(\delta_{R,s} \cdot C(R(s)) - \delta_{L,s} \cdot C(L(s))\right)ds$$
$$= \mu_0 e^{\lambda t} + \sum_i^{\#R} e^{\lambda(t-R(i))} C(R(i)) - \sum_i^{\#L} e^{\lambda(t-L(i))} C(L(i)) \tag{19}$$

$$\sigma^2(t) = \sigma_i^2 e^{\lambda t} + \frac{\sigma_a^2}{2\lambda}\left(e^{2\lambda t} - 1\right) + \int_0^t \sigma_s^2 \left(\delta_{R,s} \cdot C(R(s)) + \delta_{L,s} \cdot C(L(s))\right) e^{2\lambda t} ds$$
$$= \sigma_i^2 e^{\lambda t} + \frac{\sigma_a^2}{2\lambda}\left(e^{2\lambda t} - 1\right) + \sum_i^{\#R} \sigma_s^2 C(R(i)) e^{2\lambda(t-R(i))} + \sum_i^{\#L} \sigma_s^2 C(L(i)) e^{2\lambda(t-L(i))} \tag{20}$$

where $\#R$ is the number of right clicks on this trial up to time $t$, and $R(i)$ is the time of the $i$th right click. $C(R(i))$ tells us the effective adaptation for that clicks. For a detailed discussion of a similar model, see ref. [38]. Given a distribution of accumulation values $P(a|t, \delta_R, \delta_L) = \mathcal{N}(\mu(t), \sigma^2(t))$, and the bias parameter $B$, we can compute the left and right choice probabilities by:

$$P(\text{go right}) = \frac{1}{2}\left(1 + \text{erf}\left(\frac{-(B - \mu(t))}{\sigma\sqrt{2}}\right)\right), \tag{21}$$

$$P(\text{go left}) = 1 - P(\text{go right}). \tag{22}$$

These choice probabilities are then distorted by the lapse rate, which parameterizes how often a rat makes a random choice. The model parameters $\theta$ were fit to each rat individually by maximizing the likelihood function:

$$L = \prod_i^{\#\text{trials}} P\left(\text{rat's choice on trial } i | \theta, \delta_R^i, \delta_L^i\right). \tag{23}$$

BIC analysis supported a reduced model without the initial noise $\sigma_i$ and accumulation noise parameters $\sigma_a$ in some rats, but strongly supported keeping the parameters in other rats. Due to the presence of large discounting rates, these parameters are difficult to recover in synthetic datasets. Given the mixed BIC analysis, we included these parameters but constrained them with a half-gaussian prior on the initial noise $\sigma_i$ and accumulation noise parameters $\sigma_a$. The priors were set to match the respective best fit values from ref. [12]. Removal of these priors did not alter our conclusions about discounting strength, $\lambda$. The numerical optimization was performed in MATLAB, using the function *fmincon()*. To estimate the uncertainty on the parameter estimates, we used the inverse Hessian matrix as a parameter covariance matrix.[18] To compute the hessian of the model, we used automatic differentiation to exactly compute the local curvature[39]. See Supplementary Table 1 for parameter estimates and uncertainty values. Brunton et al.[12] extensively analyzed how well this model recovers generative parameters, finding the model contains one maximum likelihood point in parameter space (See Section 2.3.3-6 of the supplement to ref. [12]).

**Calculating noise level from model parameters**. Given the model parameters ($\sigma_s^2$, $\phi$, and $\tau_\phi$), we computed the average level of sensory adaptation on each click $\langle C\rangle$. Then, we computed what fraction of the probability mass would cross 0 to be registered as a click on the other side (See Supplementary Note 5).

$$n = \frac{1}{2}\left(1 + \text{erf}\left(\frac{-\langle C\rangle}{\sqrt{2\sigma_s^2 \langle C\rangle}}\right)\right). \tag{24}$$

**Code availability**. The code and software that support the findings of this study are available from the authors upon request.

## Data availability
The data that support the findings of this study are available from the authors upon request.

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

## Acknowledgements

We thank all members of the Brody lab for technical assistance, and feedback throughout the project. We thank Ben Scott, Diksha Gupta, Tim Hanks, and Christine Constantinople for detailed comments on the manuscript. This work was supported in part by NIH grant 5-R01-MH108358.

## Author contributions

A.P.: Task design, rat training, theoretical analysis, quantitative methods development and application. A.E.: Task design, rat training, and advised during all aspects of the study. C.B.: advised during all aspects of the study.

## Additional information

**Competing interests:** The authors declare no competing interests.

