## [Peer Review File · Nature Communications]

Reviewers' comments:

Reviewer #1 (Remarks to the Author):

The authors have extended recent experimental and theoretical work on evidence-accumulation and decision-making in dynamic environments (Glaze et al 2015) to consider a task where evidence arrives discretely, rather than continuously as in the classic random dot kinetogram. Specifically, they use a dynamic auditory clicks stimulus, and a rat must decide which ear is currently receiving Poisson clicks at a higher rate. They derive an optimal observer model, assuming the observer knows the change rate of the environment, which computes the LLR for right/left ear given a specific train of L/R Poisson clicks. When incorporating click misattribution errors analogous to those identified in Brunton et al (2013), they find that rats discount click evidence at a rate that maximizes their decision accuracy. They demonstrate this quantitatively both by performing a reverse kernel analysis and fitting a linearization of their normative model. Furthermore, they fit a more general parameterized model and find it further validates their hypothesis that rats discount at an optimal rate. They also quantify the rate that rats learn the environmental change rate by performing a sequence of many trials and dividing them into blocks with many trials each, to which they fit their models.

This work is novel and of interest to the wider community in several ways. First, it proposes a novel decision-making task that can be analyzed perhaps more easily than random dot kinetograms. Random dot kinetograms often have the issue of it being difficult to discern precisely what a subject's instantaneous collected signal-to-noise is. That problem is solved in the dynamic clicks task by using a much simpler stimulus, where possible sources of sensory noise can be more easily identified. Second, the work links a normative model to psychophysical results and proposes a theory for why subjects appear to perform suboptimally -- click misattribution. In addition, the authors have shown that a linearization, which is even easier to analyze, may be sufficient for modeling the evidence accumulation process. All in all I think this is a useful contribution.

However, I have some concerns. In particular, it appears to me that the authors have only used one set of click rates: $r_1=38\text{Hz}$ and $r_2=2\text{Hz}$, which carry a very high signal-to-noise ratio. As discussed in a more detailed comment below, I am concerned that this could have implications for model identifiability, so the authors need to explain to me why this choice was OK. Second, maybe I missed it, but I would like to see how well models with different mislocalization noises and optimal λ 's compare in terms of performance. This bears upon the broader issue of model identifiability, and I do not see much in the way of a sensitivity analysis. How confident are we that the fit λ is the only one that will work well? A few other concerns are outlined below, along with some typos I spotted. Please address all these comments in your response. As I said, I think this work is a very useful contribution, but I need more info about experimental parameter choices and model fitting.

Major Comments:

1. As far as I can see in the main text, the only choice of Poisson click rates used is $r_1 = 38\text{Hz}$ and $r_2=2\text{Hz}$. The SNR here is very high, which means that the performance of the optimal observer should be very high. Of course, I know that you are assuming that click mislocalization is occurring, which will bring down the SNR quite a bit, but I wonder why you only chose to analyze this set of click rates? When SNR is so high, it can be hard to distinguish models, because a broad range of parameters will all perform pretty well. It seems that you should have varied the SNR some, and see if your results still hold. I don't really want you to go back and run new experiments, but you need to convince me why the results for this one set of click rates is all encompassing. Can you leverage results from Brunton et al (2013)? For instance, does the click mislocalization in that work seem to be constant across click rate parameterizations?

2. The explanation you provide for suboptimal performance is click mislocalization. However, note that there is a curve of possible λ and n values that would give the same performance. How do you discern which is happening in your model fitting? It appears that all you are using to fit models is just the performance of the rat -- I guess there is nothing else that you could use. When you go to the reverse correlation analysis, you run into the same issue, right? There is a curve of λ and n values that gives the same performance, so it is not possible to tell whether you have high ' n ' and low ' λ ' or vice versa. Thus, why should I believe that ' n ' is exactly the value you have selected that conveniently yields the optimal λ in fits? You say that this is gleaned from the Brunton et al (2013) study at some points in the paper.

3. Along the same lines of my previous comment, you fit a model with more parameters in Fig. 5. Again, this identifies the rats as having high mislocalization noise and low λ . How sensitive is your model to changes in parameters? Is there a manifold of parameters that would all fit the data pretty well? You reach this strong conclusion that rats are discounting optimally, but how robust of a conclusion is this? I did not see much of an analysis of how sharp the optimal fit is. It could be that many parameter sets would yield similar performance. Please provide some results on parameter sensitivity. I did not see any in Supp. Mat. or Methods.

4. Also, a discussion of the cost function used to fit the model would help here too. Did you penalize for more parameter, using something like AIC or BIC? If not, why not? Also, did you just fit it to have the same performance as the rats did? Again, maybe it was somewhere in the manuscript, but I did not see any detailed description of the quantification used in model fitting. It seems like it would be hard for the reader to reproduce your results.

5. How are the exponential kernel fits done in Fig. 3 and otherwise? Did you simply do a least squares fit of the constant and rate of the exponential? Did you compare to the case where you added a constant? $C + A \exp(B \cdot t)$? Also, you can get different fits whether you do least squares on a linear or logarithmic scale. Which did you use?

6. I don't really understand the difference between Fig. 12 and 13 in Supp. Mat. Is there a difference between task parameters here? Performance in Fig. 13 looks more flat.

7. There is some subtlety going from Eq. (35) to (36). In particular, I am thinking that the $\delta_{t,R}$ in (35) ought to be more like a Kronecker delta, not a delta distribution, since you want it to increment a_t by some finite amount. You should allude to this. Then, the $\delta_{t,R}$ in (36) should actually be delta distributions, as you've taken the limit of $1/(\Delta t)$ as Δt goes to 0 there.

Minor Comments:

I spotted a number of typos. Please go through the document thoroughly.

- Ref to Glaze in abstract should be 2015, not 2016
- line 56: "handle over" should be "handle on"
- line 64/65: "at the moment" is stated twice in one sentence
- line 119: limit "as" Δt goes to 0
- line 348: "static stationary" redundant
- line 356: "dynamics clicks task"
- line 363: "rats"
- line 411: "subject's"
- Eq. (28): " R_{t-i} "
- Eq. (31): t appears before exponential, shouldn't be there.

-line 587: 'a' is probably not a good letter to use, since you use it to represent the LLR in the main text, unless this is the 'a' from the main text, which was not very clear to me.
-line 616: "dampen" the fluctuations

Reviewer #2 (Remarks to the Author):

It is well-reasoned to assume that sensory noise contributes to rat's decision-making behavior when performing sensory discrimination tasks in a dynamic environment. The authors address the consequences of sensory noise in a hidden state inference task and make a well-written and compelling argument that sensory noise may explain animals' less-than-maximal accuracy and the low steepness of observed evidence discounting. This paper addresses an exciting topic at the field's frontier: the treatment of evidence in hidden state inference. It represents a novel and valuable addition to computational neuroscience.

While the treatment of sensory noise is thorough, and the collected data is consistent with this explanation, we suggest that the degree of experimental testing and behavioral analysis is not sufficient for the stronger claims made in this paper. We therefore recommend three modifications. 1) We recommend that the authors refrain from strong claims about the optimality of animals' hidden state inference based on this data alone. 2) We recommend that the authors attempt to model animal's trial-by-trial performance using the full nonlinear model. We offer that this approach might be used to better differentiate the animal's estimation of environmental volatility from the animal's estimation of evidence reliability. 3) We recommend that the authors test at least one additional level of evidence reliability. In particular, we suggest an infinite r_1/r_2 click ratio. Finally, we point out that the claims in this paper would be best supported by identifying situations in which the level of sensory noise could be shown to vary (and ideally, were lower than in this task) and correspondingly predicted animals inference behavior. We believe that modifications 1) and 2) are sufficiently simple for a publication of this nature, and concede that 3) may not be necessary, but would remove much of the ambiguity about the contribution of sensory noise to the rat's behavior in this task.

Paper summary and strengths:

The authors do a beautiful job demonstrating that in the tasks they and others implement, sensory noise would necessitate less steep evidence discounting than a noiseless agent would optimally use. They not only replicate the basic finding that evidence discounting is less steep than would be expected of a maximally accurate agent, they investigate evidence discounting across three different levels of environmental volatility and find changes in the expected direction and of the expected magnitude. The authors further argue that the specific degree of evidence discounting well matches measured levels of sensory noise in this task and in a similar, previously-published task. They therefore refute the hypothesis that animals underestimate environmental volatility and instead claim that given sensory noise, evidence accumulation for hidden state inference is optimal.

Suggested modifications, in reverse order:

3) Test at least one additional level of evidence reliability.

Most of the data collected and analyzed in this paper involves a single combination of parameters: one level of external evidence reliability (38Hz/2Hz) and one level of environmental volatility (1Hz). No other evidence reliability was tested, and two other levels of environmental volatility (a trivial 0Hz and a revealing 0.5Hz) were only lightly analyzed after being tested in only 4 rats. It is far too much to claim a universal optimality of animals' hidden state inference based on this data alone. And the claim that only sensory noise contributes to the animals less-than-maximally-accurate behavior is too strong for so specific a set of tested conditions. This claim is best supported by the matching of three measured values, two in this paper (an evidence discount parameter λ , and a sensory noise

parameter n), and one in a previous work (Brunton et al., 2013). We concede these values are approximately well-matched, but argue that a replication of the same level of sensory noise for another level of external evidence reliability would greatly improve the claim.

In particular, the authors wish to discard the hypothesis that animals underestimate environmental volatility, and explain evidence discounting as only resulting from sensory noise. This could be best illustrated in a task in which underestimations of environmental volatility alone could not account for animal behavior and sensory noise could be measured more directly. With an external κ of infinity ($r_1/r_2=40\text{Hz}/0\text{Hz}$), animals would ideally only need to use the last event to infer the hidden state. In such a situation, environmental volatility would have no impact on behavior if sensory noise is near zero. However, if sensory noise does meaningfully contribute to the inference process, environmental volatility would contribute, and in a manner precisely predicted by sensory noise alone. Testing this additional level of evidence reliability would provide a good validation of the inferred level of sensory noise in this task and a useful rhetorical illustration of the insufficiency of underestimations of environmental volatility in explaining animal behavior.

2) Differentiate the animal's estimated environmental volatility from the animal's estimated evidence reliability using trial-by-trial maximum likelihood analysis of the complete, nonlinear model.

The authors analyze all of the data using a linear model. Even maximum likelihood estimations of noise and evidence discounting inferred from trial-by-trial data uses this linear model. Perhaps this analysis method is somewhat simplifying, but it requires so much deviation from the optimal manner of utilizing evidence in hidden state inference, that to rely on it so much in a paper reportedly about optimal inference is quite disheartening. To justify this approach, the authors claim that the complete nonlinear model and the linear simplification are behaviorally indistinguishable when the correct parameter is used. We are skeptical of this claim and require more illustration to accept it. In particular, equivalent levels of accuracy and reverse correlation in behavior are not sufficient to argue behavioral indistinguishability of the linear and the nonlinear model. Instead, the predicted trial-by-trial decisions must also be identical for both models. We intuitively doubt this is true and expect the authors to prove it is to accept linear simplification.

While the linear simplification may enable simpler description of animal behavior, it occludes the distinction between misestimations of environmental volatility (hazard rate h) and altered evidence discriminability (κ) due to sensory noise. By attempting to directly measure these two different parameters, h and κ , using maximum likelihood estimation from trial-by-trial data, across all three tested hazard rates, the authors could better argue that environmental volatility is correctly represented by the animal, and sensory noise causes an altered evidence discriminability and is the sole cause of less-than-maximal behavior. It would be strongest to measure these parameters across a wide combination of r_1/r_2 ratios and hazard rates to show that the same estimated sensory noise, n , explained all the data, and that the hazard rate was correctly represented in all cases. This perspective forms the basis of our recommended modification 3), described above. We recognize that these additional experiments are laborious, and therefore recommend as a minimum alternative, the extraction of these distinct parameters, h and κ , from all the data gathered so far using trial-by-trial maximum likelihood estimations. If this is not feasible, perhaps the authors could explain how they can discard the hypothesis that environmental volatility is underestimated in their task and justify that the amount of data collected so far is sufficient to claim sensory noise alone explains the observed behavior.

1) Refrain from strong claims of optimality.

Most importantly, the claim of optimality is not sufficiently justified. The optimal algorithm for hidden state inference is very specific and its utilization is not conclusively demonstrated by this paper. In other words, claims of optimal hidden state inference require a more thorough comparison of behavior than conducted here. In particular, we believe a claim of optimality requires more than showing a

predicted reverse correlation between individual events and the animal's behavior. In other words, showing that the optimal and observed behavior both have one same metric is not sufficient evidence that they are totally identical processes. A claim of optimality requires showing that the optimal inference algorithm perfectly predicts the animal's behavior. If that is not possible due to noise, it must be shown that this hidden state inference algorithm better predicts behavior than any alternative algorithm. Importantly, in hidden state inference, individual events cannot be considered independently. They dynamically modify state belief, which is updated at each timestep and depends on its previous value. Illustration of an optimal fit in reverse correlation is not sufficient evidence of optimal hidden state inference. Approximating hidden state inference using a linear model is also insufficient evidence that an optimal hidden state inference algorithm is being applied. Unless the authors can show that the trial-by-trial decision-making is well-predicted by the complete optimal algorithm (not a single simplifying parameter that doesn't describe optimal behavior), we suggest they change the title to "Sensory noise explains observed evidence discounting in a dynamic environment."

Other comments on strength of central hypothesis, utility of task design

In general, the authors wish to argue that sensory noise is a strong predictor of inference-based decision-making in commonly used psychophysical tasks. This claim would be well-made if variability in sensory noise could be shown to predictably influence the decision-making metrics this paper measures. Variability in sensory noise could be between animals, or across different sensory paradigms, or as a consequence of neural manipulation. The authors do attempt to compare individual rat's sensory noise to their evidence discounting but find no compelling relationship. Instead, they argue that the average value of sensory noise in the population approximately matches the average value of discounting in the population. To waive away the meaningfulness of the lack of a relationship, the authors point out that the observed variability in evidence discounting across rats has negligible effect on overall task accuracy.

The authors forward their task design as a good basis from which to study the neurophysiology of evidence accumulation in a dynamic environment. The authors astutely point out the great opportunity for understanding neural mechanisms for learning and representing the decision-relevant statistics of the external world. Unfortunately, this task may not be well-suited to this goal. The level of sensory noise in this task is quite high (limiting localization accuracy to only 65%), meaning that much of animal's representations and behavior is not predictable based on controlled external events. A task with much lower sensory noise would form a better basis for the desired neurophysiological investigations and would provide a good comparison point for the role of sensory noise.

Reviewer #3 (Remarks to the Author):

In this work, Piet, El Hady, and Brody investigate the ability of rats to optimally integrate point process data in the form of auditory clicks for purposes of deciding about the underlying state of a changing world. They show that when misidentification of clicks is accounted for (and estimated from a separate study), rats' ability to discount past information is optimal.

In one sense, this is an unsurprising result. That animals, especially highly overtrained ones, can learn to perform such a task optimally should not shock most readers. Nonetheless, this study is important for at least two reasons: First, in many cases, animals **can** fail to integrate information optimally, and understanding the circumstances under which this happens (or the circumstances under which behavior can be explained by constraints like noise floors) is valuable. Second, the literature on decision-making in dynamic environments has proven fruitful for our understanding of the brain's online information processing, and as the authors note, their analysis opens up new avenues of investigation for their and similar paradigms.

I enthusiastically recommend publication with only minor quibbles. I also want to congratulate the authors on what I found to be a very lucid presentation of such technical material.

Minor comments:

l. 210: It would help to elaborate on this a little more. A sentence or two recapitulating what the figure shows would help drive the point home.

Eq 15: $\Delta L, t$ should be $\Delta_{L, t}$?

Fig 5B: In the caption: isn't this pink, not orange?

Figure 6: It would be helpful if the supplement could show individuals.

Eqns 19-22: Eq 20 is just a further simplification of Eq 19, correct? Shouldn't these just be one equation with aligned right-hand side?

Eq 26: The lead-up to this is a bit terse. I managed to fill in the steps of the derivation, but it would help most readers if this was laid out a bit more explicitly. Some of this is spelled out in the supplement. A few sentences here and a pointer to the relevant sections of the supplement should do it.

Figure 20: It would be better if each plot either used the correct character (λ is rendered, but "phi" is not) or had a more informative name. I'm not sure what "ss" is.

Summary of Responses

We thank the reviewers for their detailed and very helpful comments on our manuscript. We feel that we have been able to respond to all of the issues raised. Here we provide a summary of the major changes to our manuscript, as well as a summary of major concerns from the reviewers. Below we respond to each reviewer in detail.

Summary of reviewers' concerns

Reviewer #1 had concerns about model identifiability including the robustness of our findings, the task SNR, and methodological details.

Reviewer #2 had concerns about the limited range of click rates used in our task, the linear approximation of the optimal inference process, and our claim of optimal behavior.

Reviewer #3 did not have major concerns, but did have some minor concerns which we address below by providing additional details.

Summary of major issues and changes to manuscript

(1) Concerns with strength of the claim of optimality

Both reviewers #1 and #2 raised this issue, with reviewer #2 concerned that our manuscript claimed to show that rats used optimal hidden state inference, which is difficult to disambiguate from other distinct approximations, and with reviewer #1 concerned that we had not documented how sharp the optimal fit is. Both comments are very helpful. In response, we have changed the manuscript and the title, which now reads “Rats adopt the *optimal timescale* for evidence integration in a dynamic environment” (emphasis added here). That is, following R2’s suggestion, we avoid any claims about optimal *hidden state inference*, and instead make a statement much more directly drawn from the data: the optimal *timescale* statement in the title corresponds simply to estimating the rats’ discounting rate λ and comparing it to the optimal value, as is done directly in the manuscript (Figs. 4 & 5). With regard to R1’s comment, we now clarify that the sharpness of the optimality of the rats’ timescale corresponds to confidence intervals on the estimate of λ , and now show these confidence intervals in Supp. Fig. 21, where they are shown to be relatively small.

(2) Trade-offs between different parameters, particularly between λ and “ n ”.

Reviewer #1 had two very helpful, and related, concerns regarding model-fitting. First, R1 was concerned as to how estimates of click mislocalization n might trade-off with estimates of the discount rate λ . However, as we explain below and have now clarified in the manuscript, n was not fit to our data but was instead a fixed value, taken as is from Brunton et al. 2013. Thus the estimates λ and n cannot trade off in our fits to our data. We apologize for the lack of clarity in the original submission. Second, the reviewer was concerned as to whether λ could trade off with different parameters when we fit the multi-parameter trial-by-trial model. We have now added a new supplementary figure (Supp. Fig. 28) that demonstrates that λ did not significantly trade off with any of the other parameters.

(3) Exploring other click rates.

Both Reviewer #1 and Reviewer #2 raised issues with our use of a single set of auditory click rates.

R1 was concerned that at the click rates used ($\sim 38/2$ Hz) SNR might be too high and might create model identifiability problems. We now explain that with the current click mislocalization n , variable trial durations, and environmental volatility h , we are close to the limit of overall % correct that rats are willing to tolerate before giving up on the task, which is why we did not probe harder click rates; but we also present expanded analyses to clarify and demonstrate the robustness of our model fitting. In addition, we followed R1's very helpful suggestion to examine the value of the click mislocalization parameter n for different auditory click rates in the data of Brunton et al. 2013. The results of this analysis now confirm that n is constant across auditory click rates (Supp Fig.17).

R2 very helpfully pointed out that (40/0 Hz) would be a particularly important condition to probe, for it would focus exclusively on the effects of sensory noise, because in that limit the optimal strategy is to simply use the last click, regardless of environmental volatility h , which thus becomes irrelevant. Since moving to these click rates would not reduce overall % correct, we were able to gather data at these 40/0 Hz rates. The results, now shown in Supp Fig 24, demonstrate that rats do not adopt a last click strategy, but instead discount evidence at the rate predicted by the value of sensory noise n that we used in the rest of the manuscript, thus strongly supporting our conclusions.

Together, we believe we have shown the robustness of our findings.

(4) Concerns with our linear approximation to the fully optimal nonlinear process.

Another very helpful concern from Reviewer #2 was the question of whether our linear approximation is indeed behaviorally indistinguishable from the full non-linear optimal process. R2 challenged us to prove it before the linear approximation could be considered acceptable. Following R2's comments, we have examined this in more detail, and as now described in the manuscript, we now show that if both linear and non-linear models are presented with the same trials and same realizations of noise, the two agree with each other, on an individual trial-by-trial basis, on a remarkably high 97% of trials. Moreover, no strongly distinguishing signatures are found in the remaining 3% of trials in which the two models make different predictions (Supp Fig. 19). We address this below in more detail, but in sum, we believe we have now responded to R2's challenge and documented that indeed, under our experimental conditions, the linear approximation and the nonlinear model are not behaviorally significantly different. The nonlinearity would become more important if we had lower click mislocalization rates n , which leads us to some additions to the discussion:

Additions to the Discussion. Comments from all the reviewers have helped to highlight that experimental manipulations that allowed lower click mislocalization rates n would be particularly interesting: they would allow harder click rates (see major issue (3) above), and would allow the nonlinearity to become more important, and would therefore allow distinguishing the full nonlinear theory from the simpler linear approximation (see major issue (4) above). This seems an important issue to underscore for possible following studies, and we have therefore added discussion of this point to our Discussion section, describing it as an important limitation of our study that it would be desirable to see improved in future studies.

Formatting notes:

- 1) In the following detailed responses, the reviewers concerns are in black text, and our responses are in blue text.
- 2) Figure numbers and equation numbers have changed since the original submission, our comments refer to the new numbers but reviewer comments refer to the old numbers.

Detailed Responses

Reviewer #1 (Remarks to the Author):

The authors have extended recent experimental and theoretical work on evidence-accumulation and decision-making in dynamic environments (Glaze et al 2015) to consider a task where evidence arrives discretely, rather than continuously as in the classic random dot kinetogram. Specifically, they use a dynamic auditory clicks stimulus, and a rat must decide which ear is currently receiving Poisson clicks at a higher rate. They derive an optimal observer model, assuming the observer knows the change rate of the environment, which computes the LLR for right/left ear given a specific train of L/R Poisson clicks. When incorporating click misattribution errors analogous to those identified in Brunton et al (2013), they find that rats discount click evidence at a rate that maximizes their decision accuracy. They demonstrate this quantitatively both by performing a reverse kernel analysis and fitting a linearization of their normative model. Furthermore, they fit a more general parameterized model and find it further validates their hypothesis that rats discount at an optimal rate. They also quantify the rate that rats learn the environmental change rate by performing a sequence of many trials and dividing them into blocks with many trials each, to which they fit their models.

This work is novel and of interest to the wider community in several ways. First, it proposes a novel decision-making task that can be analyzed perhaps more easily than random dot kinetograms. Random dot kinetograms often have the issue of it being difficult to discern precisely what a subject's instantaneous collected signal-to-noise is. That problem is solved in the dynamic clicks task by using a much simpler stimulus, where possible sources of sensory noise can be more easily identified. Second, the work links a normative model to psychophysical results and proposes a theory for why subjects appear to perform suboptimally -- click misattribution. In addition, the authors have shown that a linearization, which is even easier to analyze, may be sufficient for modeling the evidence accumulation process. All in all I think this is a useful contribution.

We thank the reviewer for this positive evaluation of our work.

However, I have some concerns. In particular, it appears to me that the authors have only used one set of click rates: $r_1=38\text{Hz}$ and $r_2=2\text{Hz}$, which carry a very high signal-to-noise ratio. As discussed in a more detailed comment below, I am concerned that this could have implications for model identifiability, so the authors need to explain to me why this choice was OK. Second, maybe I missed it, but I would like to see how well models with different mislocalization noises and optimal λ 's compare in terms of performance. This bears upon the broader issue of model identifiability, and I do not see much in the way of a sensitivity analysis. How confident are we that the fit λ is the only one that will work well? A few other concerns are outlined below, along with some typos I spotted. Please address all these comments in your response. As I said, I think this work is a very useful contribution, but I need more info about experimental parameter choices and model fitting.

We respond to each of these concerns in detail below.

Major Comments:

1. As far as I can see in the main text, the only choice of Poisson click rates used is $r_1 = 38\text{Hz}$ and $r_2=2\text{Hz}$. The SNR here is very high, which means that the performance of the optimal observer should be very high. Of

course, I know that you are assuming that click mislocalization is occurring, which will bring down the SNR quite a bit, but I wonder why you only chose to analyze this set of click rates?

The reviewer raises an important question as to why we picked this set of click rates, especially given that the click rates are quite different (38 to 2 Hz). The reviewer is correct that the click mislocalization brings down the SNR. However, it is important to note that the SNR and difficulty of each trial also depends heavily on the hazard rate, h , and on the time between the last state change and the end of the trial (which, with random trial durations and state changes, in many trials was quite short). As a result, these click rates are actually quite difficult. Figure 2E/F shows that in the noise regime we expect of the rats, an ideal observer can only respond correctly $\sim 77\%$ of the time. Harder click rates would decrease the maximum possible accuracy even more. We did not want the rat's accuracy to fall too low, because the rats can lose motivation and stop doing trials. We found that 38/2 Hz click rates balanced the need for difficult trials with rat motivation to perform trials. Furthermore, the normative integration timescale depends on the click rates used. Since we did not know how fast the rats could adapt their integration timescale, we did not want to have different click rates used within one behavioral session-- we wanted a well-defined integration timescale to estimate.

We modified the text to document this reasoning, adding to line 87:

Except where noted, the hazard rate $h=1\text{Hz}$, and click rates $\gamma = \log(r_1/r_2) = 3$, $r_1 \sim 38\text{ Hz}$, and $r_2 \sim 2\text{ Hz}$ were kept constant. The click rates were chosen to be as difficult as possible while keeping rat accuracy above 70%. The chosen parameter values correspond to a high difficulty; as described below, the performance of the optimal agent for these parameters is $\sim 77\%$ (Fig 2F). Trial difficulty is also heavily modulated by the duration of time since the last hidden state change, with the hardest trials being those that end shortly after a state change. Trials had random duration with random state changes. Thus, even within one set of click rates the rats performed a broad range of trial difficulties. For analytical simplicity and consistency across the task, we therefore chose to keep click rates constant in the study.

Our theory also motivates a second reason for not varying the click rates. The click reliability (K) is dominated by click mislocalization. We modified Figure 2A to show K as a function of the click rates, both with and without sensory noise. It is now easily seen that given sensory noise, the click reliability is relatively insensitive to changes in the click rates.

In addition to modifying the figure, we modified the text to document this reasoning, adding to line 208:

Figure 2A highlights that sensory noise is the dominant factor on determining click reliability. Given the average level of sensory noise, the click reliability is relatively insensitive to changing the click rates, which is another reason we did not vary the click rates used in the study.

When SNR is so high, it can be hard to distinguish models, because a broad range of parameters will all perform pretty well.

Model identifiability is a key issue in our study, and we respond in detail below in response to major concern #3. However, in the context of SNR, it is important to emphasize that we have a broad range of trial difficulties due to the random duration of trials, and the random state changes. Because trials have a unique duration of time since the last state change, we sample a large parameter space that helps to distinguish models.

It seems that you should have varied the SNR some, and see if your results still hold. I don't really want you to

go back and run new experiments, but you need to convince me why the results for this one set of click rates is all encompassing.

The reviewer makes an excellent point about the limitations of our study. It is possible that rat behavior might change in different parameter regimes. We have added a paragraph to the discussion section discussing the limitations of our current findings. The paragraph also touches on other issues discussed below.

We modified the text of the discussion to address this concern, adding:

“Our study has several potential limitations. First, the presence of sensory noise complicates the analysis of behavioral and neural data. A more direct test of our claim that sensory drives explains rats' integration timescales would be an experimental modulation of sensory noise by modifying the acoustics of the stimuli or behavioral rigs. While such a manipulation would be insightful, it falls outside the scope of this study. Lowering the sensory noise would facilitate investigations with harder click rates, and potentially distinguish the nonlinear and linear discounting models. Second, this study tested a limited range of click rates and environmental volatilities, primarily focusing on one set of parameters. Rodent behavior may deviate from the optimal timescale in different parameter regimes where other factors influence behavior. Future work should investigate how evidence integration timescales change over a broader parameter regime. Third, our study focused on the measurement of the timescale of evidence integration and discounting. In our parameter regime, the linear and nonlinear inference processes are very similar. Future work should investigate whether rodents use nonlinear discounting in a broader parameter regime.”

Can you leverage results from Brunton et al (2013)? For instance, does the click mislocalization in that work seem to be constant across click rate parameterizations?

We thank the reviewer for the excellent suggestion to examine the stability of click mislocalization in Brunton et al (2013) across different click rates. As we now explain, new analysis that we performed indicates that indeed, click localization seems to be constant across click rate parametrizations. In more detail: For each rat in the Brunton et al. study, we re-fit the behavioral model separately for each set of click rates each rat experienced. In that study, each rat performed trials with a mix of click rates in every session, so these trials are randomly interleaved. Due to limited trials, we only fit the three parameters that influence the click mislocalization calculation: Sensory noise, adaptation strength, and adaptation timescale. The other model parameters were fit to the entire dataset, and then fixed when fitting the individual click rates. While there is some variability across rats, we find no evidence of any trend across click rates. This finding strengthens our use of the average click mislocalization.

In addition, one finding from Brunton et al (2013) supports our constant click mislocalization finding. Brunton et al (2013) compared performance in trials where only one auditory click was played to predictions based on fitting their model to all other trials. If other trials with more clicks accurately estimated click mislocalization, then the model's mislocalization parameter should adequately predict error rates on those single-click trials. Indeed, this was the case (Figure 3B of Brunton, 2013). We could not repeat the analysis on our dataset, because our shortest trials were 500ms, making one-click trials extremely unlikely, but it supports our constant click mislocalization conclusion above.

We modified the text to include these findings in two places. In the main text, on Line 199:

Previous studies using the same auditory clicks have shown that rats have significant sensory noise (Brunton, 2013; Scott, 2015). We computed an estimate of the average sensory noise for the rats in Brunton, 2013, finding an average value of $n = 0.35$. Figure 2B shows κ against n and click rates r_1 and r_2 , and highlights the average rat sensory noise from Brunton, 2013. Click mislocalization for each rat was estimated by fitting a parametric model introduced below and in Brunton, 2013. See methods for estimation of click mislocalization from model parameters. Two lines of evidence suggest this level of sensory noise is reliable. First, Brunton, 2013 accurately predicted performance on single-click trials based on each rat's sensory noise (Figure 3B Brunton, 2013). Second, we found click mislocalization levels were constant across a wide range of click rates in Brunton, 2013 (Supplementary Methods, Click mislocalization is reliable).

We added this to the supplementary materials section on sensory noise

Click mislocalization is reliable across a wide range of click rates

In order to evaluate the reliability of the click mislocalization probability, we calculated it separately for different trials in Brunton 2013 based on the click rates used in the trials.

For each rat in Brunton 2013, we fit all parameters of the trial-by-trial model presented in the main text using all trials performed by the rat. In this dataset, each rat performed a mixture of trials with different click rates that were randomly interleaved. Different rats performed differently sets of click rates titrated to their performance. We then fit the model separately to each subset of trials with the same click rate. On each of these click-rate specific subsets, we fit only the three parameters that impact click mislocalization: sensory noise σ^2_s , adaptation strength ϕ , and the adaptation time constant τ_{ϕ} . For each model fit, we then computed the click mislocalization as defined in the previous section. Figure 17 shows the results. The vertical pink line shows the click rates for the dynamic task, and the horizontal line shows the value of n used in our theoretical analysis. Click mislocalization is reliable across a wide range of click rates.

Figure 17: Click mislocalization across click rates is reliable. Click mislocalization was calculated separately for trials with different click rates in Brunton, 2013. Click mislocalization is constant across a wide range of click rates.

2. The explanation you provide for suboptimal performance is click mislocalization. However, note that there is a curve of possible λ and n values that would give the same performance.

We would like to clarify there isn't a curve of possible λ and n values that would give the same performance. There is a curve of λ values that maximize accuracy given a value of n , but the accuracy (Fig 2F), and behavior (Fig 2G) are different along that curve.

How do you discern which is happening in your model fitting? It appears that all you are using to fit models is just the performance of the rat -- I guess there is nothing else that you could use. When you go to the reverse correlation analysis, you run into the same issue, right? There is a curve of λ and n values that gives the same performance, so it is not possible to tell whether you have high ' n ' and low ' λ ' or vice versa. Thus, why should I believe that ' n ' is exactly the value you have selected that conveniently yields the optimal λ in fits? You say that this is gleaned from the Brunton et al (2013) study at some points in the paper.

We apologize that in our initial submission we were not sufficiently clear that we did not fit " n " to our current data, or choose it to yield the optimal λ -- in our manuscript, " n " is a fixed value, derived from the sensory noise estimates in Brunton et al (2013) (which had a static environment within each trial). Given this single value of " n ", there is only one value of λ that would give the same performance (Fig 2G, pink lines and dot). This λ is a prediction from our theory and a measurement from Brunton et al (2013). Additionally, we can tell whether we have a high " n " - low λ or low " n " - high λ , Figure 4B shows the predictions from low and high λ . This demonstrates that our rats have λ s consistent with the prediction from our theory and data from Brunton et al (2013).

We have now modified the text to clarify that our value of " n " is a measurement from Brunton et al (2013) in several places. Most notably the paragraph added at line 199 mentioned above describing how we estimate " n ", and how reliable " n " is across click rates. In addition, in Figures 2B, 4A and 4B, we replaced the text "average rat sensory noise" with "average rat sensory noise in Brunton, 2013."

3. Along the same lines of my previous comment, you fit a model with more parameters in Fig. 5. Again, this identifies the rats as having high mislocalization noise and low λ . How sensitive is your model to changes in parameters? Is there a manifold of parameters that would all fit the data pretty well? You reach this strong conclusion that rats are discounting optimally, but how robust of a conclusion is this? I did not see much of an analysis of how sharp the optimal fit is. It could be that many parameter sets would yield similar performance. Please provide some results on parameter sensitivity. I did not see any in Supp. Mat. or Methods.

The reviewer raises important concerns about the possibility that parameters could trade off against each other, and about the sensitivity of our fits. We believe both of our primary analysis methods can firmly distinguish other behavioral strategies the rats might have used and rule-out parameter trade-offs. Our belief is based on several lines of reasoning.

(1) We added a new figure (Fig 21) to the supplement to show the confidence intervals on our exponential fits to the reverse correlation analysis. We omitted them in the main figure text for visibility of individual rats. The confidence intervals were computed by the MATLAB "fit" package used to fit the exponential.

We modified the methods section:

"All exponential fits were computed using the MATLAB package 'fit,' which used a least squares fit on a linear scale. 95% Confidence intervals on the exponential fits are shown in the supplement and

calculated by the `fit` package.”

We modified the supplemental section “Psychophysical reverse correlation details” to include this figure:

Figure 21. Reverse Correlation Timescales with uncertainty. Same analysis as Figure 3B, but with 95% confidence intervals on the exponential time constant fits.

(2) Figure 25 (and Table 1) in the supplement show parameter uncertainty for each parameter in the trial-by-trial model. These parameter uncertainties are derived from using the Hessian of the likelihood function as a parameter covariance matrix (Daw, 2011). We see from this figure that our uncertainty around evidence discounting (λ) is well-constrained.

(3) The reviewer’s concern about a manifold of parameters in the trial-by-trial model that could fit the data well would indicate a trade-off between parameters. To investigate the presence of trade-offs in our parameters, we examined the eigenvector decomposition of the Hessian matrix at each rat’s maximum likelihood estimate. A trade-off would appear in this analysis as an eigenvector that has significant projections onto two or more parameters (Daw, 2011). Looking across all of our rats, we find no rats with parameter trade-offs involving the evidence discounting parameter λ (new Figure 28). For each rat we find exactly one eigenvector tightly aligned with the evidence discounting parameter, and no other parameters.

(4) Residual analysis of the trial-by-trial model errors does not show any systematic pattern. For each rat we plotted how well the model fit each trial as a function of trial duration. We find these curves to be flat over time, indicating that the model fits short and long trials equally well (Figures 26 and 27).

We modified the text to document points 2-4.

In the results section:

“To evaluate parameter sensitivity in our model, we approximated the local likelihood landscape by the Hessian matrix. The inverse of the hessian matrix was then used as an estimate of the parameter covariance (Daw, 2011). Table 1 in the supplement shows parameter values and parameter uncertainty for each rat. We used the eigenvalue decomposition of each rat’s Hessian matrix to assess whether parameters in our model trade off against each other. Eigenvectors significantly aligned with multiple parameters can indicate trade-offs in the likelihood landscape. We found no significant trade-offs involving the discounting parameter λ (Fig. 28). Finally, we plotted the residual error plots for

each rat to identify systematic errors by the model. For each rat, the residual error was constant in time, indicating our model fit short and long duration trials equally well (Figs 26 & 27)."

In the methods section, we added the following sentence: *"See the Supplemental Materials for parameter estimates and uncertainty values."*

In the supplementary materials, we added some brief text at the beginning of the model details section organizing the figures:

"In this section we provide additional analysis on our trial-by-trial model. First, we provide parameter estimates and parameter uncertainty values for each rat, and compare them to rats from Brunton, 2013 (Fig 25, Table 1). Second, we plot model residual error against time (Fig 26/27). Third, we show an analysis which rules out significant parameter trade-offs involving the evidence discounting parameter λ (Fig 28)."

We also added Figure 28 to the Supplementary Materials which show the parameter trade-off analysis for each rat.

Figure 28. **No parameter trade-offs involving evidence discounting λ .** Eigenvector decomposition of Hessian approximation of likelihood landscape at best fit parameters demonstrates no significant parameter trade-offs involving the evidence discounting parameter λ . For each rat, a histogram of eigenvector weights on λ . Each rat has one eigenvector tightly aligned with λ . The vertical dashed line indicates the weight that would result from an eigenvector that projected equally on to all parameters. Parameter trade-offs would appear as an eigenvector with weight greater than the dashed line, and less than one.

(5) The rate of click mislocalization “n” is well constrained in our model fits. Figure 5C shows our uncertainty on “n” for each rat. Because “n” is a nonlinear function of several parameters, we measured the uncertainty by sampling each parameter value according to the Hessian-derived covariance matrix. For each sample of parameters, we computed “n” as described in the methods

We modified the text in the Supplemental materials, section “Click Mislocalization from Model Parameters”:

“To estimate uncertainty on n due to uncertainty on the underlying model parameters, we performed a bootstrapping analysis. Each parameter was sampled 100 times according to the Hessian-derived covariance matrix. Each parameter sample was used to generate a sample n as described above. Figure 5C shows the maximum likelihood estimate of n and its standard error.”

(6) This model has already been extensively analyzed (Section 2.3.3-6 of the Supplement for Brunton, 2013). Importantly, Brunton, 2013 found that model parameters can be reliably recovered, and the model contains one maximum likelihood point in parameter space. We therefore feel confident in relying on our model fits.

We modified the text to include this point:

“Brunton, 2013 extensively analyzed how well this model recovers generative parameters, finding the model contains one maximum likelihood point in parameter space (See Section 2.3.3-6 of the supplement to Brunton, 2013).”

4. Also, a discussion of the cost function used to fit the model would help here too. Did you penalize for more parameter, using something like AIC or BIC? If not, why not? Also, did you just fit it to have the same performance as the rats did? Again, maybe it was somewhere in the manuscript, but I did not see any detailed description of the quantification used in model fitting. It seems like it would be hard for the reader to reproduce your results.

The reviewer raises important concerns about our methods for fitting the trial-by-trial model. We apologize our initial submission was not sufficiently clear. Details about our model’s cost function and model fitting are in the methods section “Behavioral Model.” This section describes how we compute choice probabilities from the model, and how those choice probabilities are used to evaluate the likelihood of observing the rat’s choices. To help future readers, we directed them to this methods section from the results, modifying this sentence on l.296: *“The model was fit to individual rats by maximizing the likelihood of observing the rat’s choice on each trial (See Methods for details).”*

We did not penalize for more parameters using AIC or BIC when fitting the model because we were not selecting between multiple models. However, we did use BIC to compare the full model to reduced models without certain parameters. A BIC analysis comparing the full model with reduced models without initial noise and accumulation noise favored removing those parameters in some rats, but not all rats. Given the mixed results across rats, we adopted to constrain these parameters with a half-gaussian prior based on the population average from Brunton, 2013. If we remove these parameters or the priors, the findings about λ and n do not change.

We documented this in the methods section:

“BIC analysis supported a reduced model without the initial noise (σ_i) and accumulation noise parameters (σ_a) in some rats, but strongly supported keeping the parameters in other rats. Due to the presence of large discounting rates, these parameters are difficult to recover in synthetic datasets. Given the mixed BIC analysis, we included these parameters but constrained them with a half-gaussian prior on the initial noise (σ_i) and accumulation noise parameters (σ_a). The priors were set to match the respective best fit values from Brunton, 2013. Removal of these priors did not alter our conclusions about discounting strength, λ . The numerical optimization was performed in MATLAB, using the function “fmincon.” To estimate the uncertainty on the parameter

estimates, we used the inverse hessian matrix as a parameter covariance matrix (Daw, 2011) . To compute the hessian of the model, we used automatic differentiation to exactly compute the local curvature (Revels, 2016). See the Supplemental Materials for parameter estimates and uncertainty values. Brunton, 2013 extensively analyzed how well this model recovers generative parameters, finding the model contains one maximum likelihood point in parameter space (See Section 2.3.3-6 of the supplement to Brunton, 2013)."

5. How are the exponential kernel fits done in Fig. 3 and otherwise? Did you simply do a least squares fit of the constant and rate of the exponential? Did you compare to the case where you added a constant? $C+A*\exp(B*t)$? Also, you can get different fits whether you do least squares on a linear or logarithmic scale. Which did you use?

We thank the reviewer for pointing out the lack of details. We have amended the text in the main methods to include more detail about the exponential fits:

"The excess rate curves were then normalized to integrate to one. This was done to remove distorting effects of a lapse rate, as well to make the curves more interpretable by putting the units into effective weight of each click on choice. To quantify the timescale of the reverse correlation curves, we fit an exponential of the form $ae^{(bt)}$ to each curve. The parameter a is a scale parameter. The parameter b is the discounting rate, while $1/b$ is the integration timescale. All exponential fits were computed using the MATLAB package "fit," which used a least squares fit on a linear scale."

We did not perform log-level regression because the reverse correlation curves contain negative and zero-valued data points. We did not compare our fits to the case with a baseline constant "C" because the reverse correlation curves go to zero at long timescales.

6. I don't really understand the difference between Fig. 12 and 13 in Supp. Mat. Is there a difference between task parameters here? Performance in Fig. 13 looks more flat.

We apologize for the lack of clarity. Both figures show the same data, but organized in a different manner. Figures 11/12 show chronometric plots for each rat with respect to the duration of the final state. Figures 13/14 show chronometric plots for each rat with respect to the total duration of the trial. We have modified the labels for each figure to better reflect this distinction. Both sets of figures were previously labeled "*Chronometric graph for all rats*." Now Figures 11/12 are labeled "*Chronometric graph for all rats with respect to final state duration*", and Figures 13/14 are labeled "*Chronometric graph for all rats with respect to total trial duration*."

7. There is some subtlety going from Eq. (35) to (36). In particular, I am thinking that the $\delta_{t,R}$ in (35) ought to be more like a Kronecker delta, not a delta distribution, since you want it to increment a_t by some finite amount. You should allude to this. Then, the $\delta_{t,R}$ in (36) should actually be delta distributions, as you've taken the limit of $1/(\Delta t)$ as Δt goes to 0 there.

We thank the reviewer for this astute observation and for pointing out the potential for confusion. Note the equation numbers changed due to changes earlier in the manuscript. We have modified the text around Eq. 35 (now Eq.34) and 36 (now Eq. 35) to reflect our interpretation of the delta functions in each case. As the reviewer suggests, the important idea is that both equations have an increment of value 1 for each delta. In the discrete case (Eq. 34), this is interpreted as a Kronecker delta that has amplitude 1. In the continuous case (Eq. 35), this is interpreted as a Dirac delta that integrates to 1.

We modified the text to read :

“Care needs to be used when interpreting the δ terms in Eq. 34 and Eq. 35. In both cases we want each δ to increment (or decrement) the accumulation variable by a value of 1. In Eq. 34, this leads to interpreting the δ terms as Kronecker Deltas that have amplitude 1 when $R/L=t$. In Eq. 35, this leads to interpreting the δ terms as Dirac Deltas that integrate to 1 when $R/L=t$.”

In addition, we added an intermediate step to the derivation before what is now Eq 34. This step is presented in the main text, but we realized it added readability to repeat it here.

*“Plugging in the evaluation of the log-evidence term from the main text:
 $da = (\delta_r - \delta_l) * k - 2h * dt * \sinh(a)$.”*

Minor Comments:

I spotted a number of typos. Please go through the document thoroughly.

We thank the reviewer for pointing out these typos, they have been corrected without comment unless noted individually below. (Note that line numbers in the document have changed)

-Ref to Glaze in abstract should be 2015, not 2016

-line 56: “handle over” should be “handle on”

-line 64/65: “at the moment” is stated twice in one sentence

-line 119: limit “as” Δt goes to 0

-line 348: “static stationary” redundant

-line 356: “dynamics clicks task”

-line 363: “rats”

-line 411: “subject’s”

Eq. (28): “ R_{t-i} ”

Eq. (31): t appears before exponential, shouldn’t be there.

-line 587: ‘a’ is probably not a good letter to use, since you use it to represent the LLR in the main text, unless this is the ‘a’ from the main text, which was not very clear to me.

We thank the reviewer for pointing out this ambiguity, we have changed the variable “a” in this section.

-line 616: “dampen” the fluctuations

Reviewer #2 (Remarks to the Author):

It is well-reasoned to assume that sensory noise contributes to rat’s decision-making behavior when performing sensory discrimination tasks in a dynamic environment. The authors address the consequences of sensory noise in a hidden state inference task and make a well-written and compelling argument that sensory noise may explain animals’ less-than-maximal accuracy and the low steepness of observed evidence discounting. This paper addresses an exciting topic at the field’s frontier: the treatment of evidence in hidden state inference. It represents a novel and valuable addition to computational neuroscience.

We thank the reviewer for this positive evaluation of our work.

While the treatment of sensory noise is thorough, and the collected data is consistent with this explanation, we suggest that the degree of experimental testing and behavioral analysis is not sufficient for the stronger claims made in this paper. We therefore recommend three modifications. 1) We recommend that the authors refrain from strong claims about the optimality of animals' hidden state inference based on this data alone. 2) We recommend that the authors attempt to model animal's trial-by-trial performance using the full nonlinear model. We offer that this approach might be used to better differentiate the animal's estimation of environmental volatility from the animal's estimation of evidence reliability. 3) We recommend that the authors test at least one additional level of evidence reliability. In particular, we suggest an infinite r_1/r_2 click ratio. Finally, we point out that the claims in this paper would be best supported by identifying situations in which the level of sensory noise could be shown to vary (and ideally, were lower than in this task) and correspondingly predicted animals inference behavior. We believe that modifications 1) and 2) are sufficiently simple for a publication of this nature, and concede that 3) may not be necessary, but would remove much of the ambiguity about the contribution of sensory noise to the rat's behavior in this task.

We respond to each of these concerns in detail below.

Paper summary and strengths:

The authors do a beautiful job demonstrating that in the tasks they and others implement, sensory noise would necessitate less steep evidence discounting than a noiseless agent would optimally use. They not only replicate the basic finding that evidence discounting is less steep than would be expected of a maximally accurate agent, they investigate evidence discounting across three different levels of environmental volatility and find changes in the expected direction and of the expected magnitude. The authors further argue that the specific degree of evidence discounting well matches measured levels of sensory noise in this task and in a similar, previously-published task. They therefore refute the hypothesis that animals underestimate environmental volatility and instead claim that given sensory noise, evidence accumulation for hidden state inference is optimal.

We thank the reviewer for this positive evaluation of our work.

Suggested modifications, in reverse order:

3) Test at least one additional level of evidence reliability.

Most of the data collected and analyzed in this paper involves a single combination of parameters: one level of external evidence reliability (38Hz/2Hz) and one level of environmental volatility (1Hz). No other evidence reliability was tested, and two other levels of environmental volatility (a trivial 0Hz and a revealing 0.5Hz) were only lightly analyzed after being tested in only 4 rats. It is far too much to claim a universal optimality of animals' hidden state inference based on this data alone. And the claim that only sensory noise contributes to the animals less-than-maximally-accurate behavior is too strong for so specific a set of tested conditions. This claim is best supported by the matching of three measured values, two in this paper (an evidence discount parameter λ , and a sensory noise parameter n), and one in a previous work (Brunton et al., 2013). We concede these values are approximately well-matched, but argue that a replication of the same level of sensory noise for another level of external evidence reliability would greatly improve the claim.

Before addressing the broader question of an additional level of evidence reliability, we would like to point to the new analysis introduced above demonstrating that the sensory noise levels found in the Brunton et al., 2013 data are constant across a broad range of click rates. We feel this analysis speaks to the generalizability of our current findings.

In particular, the authors wish to discard the hypothesis that animals underestimate environmental volatility,

and explain evidence discounting as only resulting from sensory noise. This could be best illustrated in a task in which underestimations of environmental volatility alone could not account for animal behavior and sensory noise could be measured more directly. With an external kappa of infinity ($r_1/r_2=40\text{Hz}/0\text{Hz}$), animals would ideally only need to use the last event to infer the hidden state. In such a situation, environmental volatility would have no impact on behavior if sensory noise is near zero. However, if sensory noise does meaningful contribute to the inference process, environmental volatility would contribute, and in a manner precisely predicted by sensory noise alone. Testing this additional level of evidence reliability would provide a good validation of the inferred level of sensory noise in this task and a useful rhetorical illustration of the insufficiency of underestimations of environmental volatility in explaining animal behavior.

We thank the reviewer for this insightful suggestion to look at rat behavior in a high reliability environment (infinite click reliability κ in the case of no sensory noise). The reviewer is correct that this directly tests whether poor estimates of environmental volatility could explain the observed discounting rates in our rats. We trained two rats on trials with high click reliability ($r_1/r_2 = 40\text{Hz}/0\text{Hz}$), if the observer has no sensory noise, this results in an infinite κ ($K = \log(40/0)$). An infinite κ means the ideal observer should base its decision purely on the last click played. Crucially, as the reviewer explained, the environmental volatility estimate becomes irrelevant to optimal behavior. However, if the observer has sensory noise, then the observer should discount evidence as derived in the main text. The expected discounting rate with sensory noise is very similar to the expected rate on the standard click rates, because the click reliability does not change very much due to large sensory noise (See new pink line in Figure 2A).

Two rats performed trials in a high reliability environment and Figure 24 shows the behavior of the two rats quantified by reverse correlation and model fits. We find the rats do not adopt a last click strategy, consistent with sensory noise driving the observed discounting rates, not poor estimates of the environmental volatility. Optimal agents without sensory noise would base their decision purely on the last click, and would appear in our reverse correlation analysis and model fits as discounting agents with discounting rates at 40 clks/s.

We modified the text, adding in the discussion:

“Our linear approximation does not parameterize environmental volatility, so we did not estimate our rat's estimate of this parameter. However, with the estimate of sensory noise from Brunton, 2013 we could accurately predict the rat's integration timescales without considering environmental volatility. Figure 24 directly examines whether poor estimates of environmental volatility could explain rat behavior. We find that in a high click reliability environment, where environmental volatility estimates should not influence integration timescales, rats still discount consistent with the sensory noise predicted timescale. “

In the supplement we added a section describing this experiment:

Ruling out poor estimates of environmental volatility

Glaze, 2015 examined human discounting behavior on a comparable task, finding that human subjects failed to discount at the optimal rate due to poor estimates of the environmental volatility. Our linear approximation to the full inference process does not directly parameterize environmental volatility. In order to evaluate whether our rat's discounting rates could be explained by a poor estimate of environmental volatility rather than sensory noise, we trained two rats on trials with $r_1 = 40$ hz, and $r_2 = 0$ hz. If the rats had no sensory noise, then the reliability of each click at informing the current state becomes infinite $\kappa(r_1, r_2) = \log(r_1/r_2) = \infty$. An infinite click reliability means the rats could achieve perfect accuracy by choosing the side based on the last click played. Crucially, with an

infinite click reliability, the estimate of the environmental volatility becomes irrelevant. If the observed discounting rates in the main text were driven by poor estimates of the environmental volatility and not sensory noise, then in this high reliability environment the rats would make their choices based on the last click. However, if sensory noise is present, then the theory presented in the main text predicts the rats should discounting evidence at a rate comparable to the main results. Figure 24A shows the reverse correlation discounting rates, comparable to Figure 4B. In addition, the discounting parameter for the model fit to these rats is displayed. We see the rats do not adopt a last-click strategy, consistent with sensory noise being the primary driver of the rat's discounting rates. Figure 24B shows the session by session accuracy for each rat, showing no trend of increasing accuracy or adaptation to the high click reliability environment.

Figure 24. Rats in a high click reliability environment behave consistently with high sensory noise, not poor volatility estimation. (A) Quantification of discounting timescales for rats in a high reliability environment. Reverse correlation integration timescales for rats in a high reliability environment ($r_1 = 40$, $r_2 = 0$), compared to optimal linear models with and without sensory noise. Model discounting parameters are also plotted. The horizontal dashed line shows the discounting rate expected from a last-click agent. (B) Accuracy for each behavioral session for two rats. Dotted line is average accuracy across all sessions. No apparent trend in increasing accuracy to suggest the rats are adapting to the high click reliability environment.

2) Differentiate the animal's estimated environmental volatility from the animal's estimated evidence reliability using trial-by-trial maximum likelihood analysis of the complete, nonlinear model.

The authors analyze all of the data using a linear model. Even maximum likelihood estimations of noise and evidence discounting inferred from trial-by-trial data uses this linear model. Perhaps this analysis method is somewhat simplifying, but it requires so much deviation from the optimal manner of utilizing evidence in hidden state inference, that to rely on it so much in a paper reportedly about optimal inference is quite disheartening. To justify this approach, the authors claim that the complete nonlinear model and the linear simplification are behaviorally indistinguishable when the correct parameter is used. We are skeptical of this claim and require more illustration to accept it. In particular, equivalent levels of accuracy and reverse correlation in behavior are not sufficient to argue behavioral indistinguishability of the linear and the nonlinear model. Instead, the predicted trial-by-trial decisions must also be identical for both models. We intuitively doubt this is true and expect the authors to prove it is to accept linear simplification.

The reviewer raises important concerns about our use of a linearized model. First, we present a better motivation for the use of the linear approximation, which is based on three points. The linear model has an analytical solution; is very accurate; and we expect the hazard parameter h and reliability parameter k to trade-off against each other. Then, we demonstrate that both models make comparable choices on a trial-by-trial basis. Finally, we document our changes to the manuscript to reflect our reasoning and respond to the reviewer's concern.

Motivation for use of linear model

First, the linear model has an analytical solution, so the entire moment-by-moment distribution of possible accumulation values can be solved exactly. An analytical solution to the nonlinear model is outside of the scope of the current study. In order to fit the nonlinear model, we would need to use numerical methods for nonlinear partial differential equations.

Second, we want to underscore how accurate the linear approximation is in our parameter regime. In the original manuscript we showed Figure 2E/F which demonstrate that the accuracy is very close, but we did not quantify that accuracy. We find the linear parameter by numerically optimizing for accuracy on a training set. Evaluated on a test set, the linear approximation has 99.8% of the accuracy of the nonlinear model (77.15%, and 77.34% accuracy). The training and test sets have different realizations of noise added to the clicks, with the same noise realization used for the linear and nonlinear models. We therefore conclude the linear approximation to be a very near optimal model. The nonlinear model produces an additional correct choice on one out of every 500 trials. In the context of rat behavior, optimal nonlinear behavior would result in one additional reward every 1-2 days of training over optimal linear behavior. We do not believe rats are sensitive to such small changes in reward.

Third, and most importantly, we expect the hazard parameter h and the click reliability parameter k would be very difficult to distinguish in our parameter regime. We find that across a broad range of click rates, hazard rates, and sensory noise levels, the linear approximation is accurate. The high level of accuracy of the linear approximation suggests that the main parameter we can recover is the combination of h and k that is captured by the linear discounting rate λ . We therefore expect that fitting a nonlinear model would produce estimates of h and k that trade off against each other.

We therefore respectfully disagree with the reviewer's suggestion of fitting the nonlinear model. We believe that a numerical implementation of the nonlinear model lies outside the scope of the current study, and would have relatively little value. Instead, we believe we can demonstrate that the linear model satisfies the reviewer's concerns. Namely, that the model makes the same choices on a trial-by-trial basis, and that the models are difficult to distinguish.

Trial-by-trial agreement between linear and nonlinear models

The reviewer correctly points out that we demonstrated equivalent overall accuracy between the two models, but did not demonstrate trial-by-trial identical choices between the two models. Here, we provide that analysis. When calculating consistency between the models, the role of noise needs to be considered carefully. We simulated the linear and nonlinear models 20 times on one set of 30k trials, but each simulation used a different realization of noise on the click trains. We then computed the average agreement between each pair of simulations. The linear model was self-consistent on average across noise realizations on 71.98% of trials. The nonlinear model was self-consistent on average across noise realizations on 72.87% of trials. The linear and nonlinear models were cross-consistent on average across noise realizations on 72.40% of trials. The models were cross-consistent on the same noise realizations on 97.2% of trials. Importantly, when analyzing rat behavior, we do not have access to the noise realization experienced by the rat. Given the equivalent levels

of self-consistency for each model to the cross-consistency between the models, we conclude the models would be difficult to distinguish. Most importantly, we also conclude the models would have equivalent predictive power when generalizing to new data from the rats. With the additional consideration that the linear model has one less parameter than the nonlinear model (λ vs h and κ), we find support for the use of the linear model.

Given that the cross-consistent on the same noise realization is not exact (97.2% vs 100%), along with the comparable overall accuracy, means the linear model sometimes makes correct choices when the nonlinear model makes mistakes and vice-versa. To address the reviewer's concerns about similar trial-by-trial choices, we wanted to investigate the trials where the models make different choices. Difficulty on this task is determined by the time since the last hidden state change. The ~3% of trials where the models disagree are mostly evenly spread out in time since the last hidden state change (figure below). Because the disagreements are spread out in time, there is not an obvious signature of linear vs nonlinear discounting we could examine in the rat behavior.

Taken together, we believe the linear model is a suitable approximation to the full nonlinear theory given the analysis methods we applied to our data. Importantly, we think it satisfies the reviewer's concerns about trial-by-trial choices. In addition, as discussed below in response to major concern #1, we have modified our claims and manuscript title to address any outstanding concerns about the linear approximation, and better communicate that our claims are based around a linear model.

Changes to manuscript to address the use of the linear model

We modified the text in several places to clarify our motivation for using the linear approximation, and to address the issues mentioned by the reviewer. In several places in the text we used phrases like "indistinguishable" to describe the linear vs nonlinear models. This language overstates the approximation, and we have rephrased these statements to better described the very accurate but not identical models.

In the results section "Linear approximation to nonlinear discounting" we added:

"For the average level of sensory noise, we find the linear agent to have 99.8% of the accuracy of the nonlinear model. The linear model was optimized on a training set of trials, and both models were evaluated on a test set of held-out trials and achieved 77.15% and 77.34% accuracy respectively."

And in the next paragraph we modified the language:

"We did not examine whether rats demonstrate nonlinear evidence discounting because the linear approximation matches the accuracy of the nonlinear theory. See the Supplementary Materials for more information on the use of the linear approximation."

In the results section about the model, we added:

"We parameterized the model with linear discounting, rather than nonlinear discounting in the full theory for three reasons. First, the linear discounting model has been fit to rat behavior in static environments, allowing a direct comparison to previous results. Second, the linear model has an analytical solution that greatly facilitates analysis (See methods). Third, the linear model has comparable accuracy to the nonlinear model with less parameters, simplifying the fitting procedure and providing a more parsimonious description of rat behavior."

In the discussion section on the potential limitations of our study, we added:

“Third, our study focused on the measurement of the timescale of evidence integration and discounting. In our parameter regime, the linear and nonlinear inference processes are very similar. Future work should investigate whether rodents use nonlinear discounting in a broader parameter regime.”

In the supplementary materials we added a section “**Linear Approximation Details**”

“We performed three analyses in order to demonstrate that the linear approximation is difficult to distinguish from the full nonlinear inference process on a trial-by-trial basis. First, we compared the cross-consistency between models to the self-consistency of each model across repeated simulation of the same trials with different sensory noise realizations. The nonlinear inference process, and the linear approximation were simulated on a dataset of 30,000 trials. The simulations were run 20 times with different noise realizations on each simulation. We calculated on average how self-consistent the linear (71.98% +/- 0.48) and nonlinear (72.87% +/- 0.52) models were across noise realization. In addition, we calculated the cross-consistency between the linear and nonlinear models when the noise realization was different (72.40% +/- 0.50), or the same (96.98% +/- 0.18). The value reported is the average percentage agreement between all possible pairs in each group with 95% confidence intervals. Given that the cross-consistency between models is the same as the self-consistency for each model, we conclude the two models are hard to distinguish, and have similar predictive ability on new trials. Importantly, the linear model achieves similar behavior with one less parameter.

Our second analysis looked at the small set (3%) of trials where the nonlinear and linear models disagreed when given the same noise realization. Trial difficulty on our task is determined by the length of time since the last hidden state switch. We found that the two models disagreed slightly more often immediately after a state transition, but overall disagreements were spread across all trials (Fig 19C). We then computed the chronometric curves which show model accuracy as a function of time since the last hidden state switch. We find the two models have overlapping confidence intervals at all time points (Fig 19B). We conclude it would be difficult to distinguish these two models on this dataset.

Finally, to provide intuition as to how a linear function could accurately approximate a nonlinear function we computed the distribution of values the nonlinear model would produce after being simulated on a large dataset of trials (Figure 20). The discounting term bounds how much evidence can accumulate for either of the two states. This creates a distribution of evidence values over a narrow domain. Comparing this distribution with the linear and nonlinear discounting functions, we see the linear approximate is close to the nonlinear function in the same domain as the distribution of accumulation values. This provides intuition as to how the linear function is able to approximate the nonlinear discounting function.”

In the supplement we added Figure 19, with this caption:

Figure 19. **Linear approximation is hard to distinguish from the full nonlinear inference.** (A) Average pairwise agreement across repeated simulations of the same trials with different noise realizations for the linear (L) and nonlinear (N) models. Panel shows the self consistency for the linear (L/L) and nonlinear (N/N) models, as well as the cross-consistency with the same noise (L/N same), and different noise (L/N diff.). (B) Chronometric plot shows accuracy for both models as a function of time since the last hidden state switch, simulated on a dataset of 30,000 trials with the same noise realization. Shaded bands show 95% confidence intervals. The two models have the same pattern of accuracy over time. (C) Same simulations as in B, but showing the percentage of trials where the two models disagree over time. The dotted line shows the overall percentage of trials where the models disagree. The models disagree slightly more often immediately following state changes, but consistently disagree at the same rate across time.”

In the supplement we added Figure 20, with this caption:

Figure 20. **Linear approximation is accurate in the domain of distribution of accumulation values.** To provide intuition as to how a linear function could accurately approximate the highly nonlinear discounting function we compare the distribution of accumulation values to the domain of the discounting functions. (Top) the distribution of accumulation value, a , from simulating the full nonlinear discounting theory on a dataset of trials. (Bottom) The full nonlinear and best linear approximation of the discounting functions. The discounting function bounds how far away from zero the evidence can

accumulate. The linear approximation is close to the nonlinear function in precisely the same domain as the distribution of accumulated evidence.

While the linear simplification may enable simpler description of animal behavior, it occludes the distinction between misestimations of environmental volatility (hazard rate h) and altered evidence discriminability (κ) due to sensory noise. By attempting to directly measure these two different parameters, h and κ , using maximum likelihood estimation from trial-by-trial data, across all three tested hazard rates, the authors could better argue that environmental volatility is correctly represented by the animal, and sensory noise causes an altered evidence discriminability and is the sole cause of less-than-maximal behavior. It would be strongest to measure these parameters across a wide combination of r_1/r_2 ratios and hazard rates to show that the same estimated sensory noise, n , explained all the data, and that the hazard rate was correctly represented in all cases. This perspective forms the basis of our recommended modification 3), described above. We recognize that these additional experiments are laborious, and therefore recommend as a minimum alternative, the extraction of these distinct parameters, h and κ , from all the data gathered so far using trial-by-trial maximum likelihood estimations. If this is not feasible, perhaps the authors could explain how they can discard the hypothesis that environmental volatility is underestimated in their task and justify that the amount of data collected so far is sufficient to claim sensory noise alone explains the observed behavior.

The reviewer raises an important question as to whether environmental volatility may explain our results. Our new results in a high click-reliability environment demonstrate that sensory noise drives integration timescales in a setting where miss-estimation of environmental volatility would not be sufficient to change behavior. However, we did not directly estimate or measure the rat's environmental volatility. We have extended the discussion to provide some comments on how our findings are both similar and dissimilar to the findings of Glaze, 2015 which found that miss-estimation of volatility explained human behavior. Our principal reasoning for discarding the volatility hypothesis in our data is that we can predict the rat's integration timescales by only considering sensory noise.

Glaze, 2015 examined human decision making in a dynamic environment, and found that that humans show nonlinear evidence discounting, but do not match the optimal inference process. Quantifying the subject's estimates of environmental volatility, rather than the discounting rate parameterized here, they found that subjects typically underestimated the volatility (in a 2 Hz environment comparable to our task; Figure 7 of Glaze, 2015). This finding is consistent with the role of sensory noise decreasing the discounting rate. Incorporating models of human sensory noise into their analysis could potentially explain deviations from optimality in their data. However, our rats performed a much larger number of trials than the human subjects in Glaze, 2015. Human subjects with more experience may more closely match optimal processes. Unlike Glaze, 2015, we did not examine whether our subjects demonstrated nonlinear evidence discounting because the linear approximation in our task is very close to the full nonlinear theory (Figure 2). Our linear approximation does not parameterize environmental volatility, so we did not estimate our rat's estimate of this parameter. However, with the estimate of sensory noise from Brunton, 2013 we could accurately predict the rat's integration timescales without considering environmental volatility.

1) Refrain from strong claims of optimality.

Most importantly, the claim of optimality is not sufficiently justified. The optimal algorithm for hidden state inference is very specific and its utilization is not conclusively demonstrated by this paper. In other words, claims of optimal hidden state inference require a more thorough comparison of behavior than conducted here. In particular, we believe a claim of optimality requires more than showing a predicted reverse correlation between individual events and the animal's behavior. In other words, showing that the optimal and observed

behavior both have one same metric is not sufficient evidence that they are totally identical processes. A claim of optimality requires showing that the optimal inference algorithm perfectly predicts the animal's behavior. If that is not possible due to noise, it must be shown that this hidden state inference algorithm better predicts behavior than any alternative algorithm. Importantly, in hidden state inference, individual events cannot be considered independently. They dynamically modify state belief, which is updated at each timestep and depends on its previous value. Illustration of an optimal fit in reverse correlation is not sufficient evidence of optimal hidden state inference. Approximating hidden state inference using a linear model is also insufficient evidence that an optimal hidden state inference algorithm is being applied. Unless the authors can show that the trial-by-trial decision-making is well-predicted by the complete optimal algorithm (not a single simplifying parameter that doesn't describe optimal behavior), we suggest they change the title to "Sensory noise explains observed evidence discounting in a dynamic environment."

The reviewer raises an important concern about our intended claims about rat behavior. The reviewer raises a distinction between our analysis and the nonlinear inference process. Our analysis is based around the linearized model, and while we have shown the linear model to be both highly accurate (99.8% of nonlinear accuracy) and very consistent with the nonlinear model (97% agreement with nonlinear model), the reviewer is correct that we did not analyze nonlinear discounting. Therefore, we have modified our language in the manuscript, referring to claims of optimality by "*optimal timescale*" or "*optimal linear*." In addition, we changed the title of the manuscript to "*Rats adopt the optimal timescale for evidence integration in a dynamic environment*." The term optimal timescale is a more direct communication of our analysis and our conclusions by describing the linearization used in our analysis. In addition, we this title addresses the reviewer's concern about the linear approximation to the optimal inference process. We are willing to consider other titles and phrasing of our claims if the reviewer is still concerned about our interpretation of our data.

Finally, we added a statement to the discussion about the limitations of our current work and nonlinear discounting:

"Our study has several potential limitations. ... Third, our study focused on the measurement of the timescale of evidence integration and discounting. In our parameter regime, the linear and nonlinear inference processes are very similar. Future work should investigate whether rodents use nonlinear discounting in a broader parameter regime. "

Other comments on strength of central hypothesis, utility of task design

In general, the authors wish to argue that sensory noise is a strong predictor of inference-based decision-making in commonly used psychophysical tasks. This claim would be well-made if variability in sensory noise could be shown to predictably influence the decision-making metrics this paper measures. Variability in sensory noise could be between animals, or across different sensory paradigms, or as a consequence of neural manipulation. The authors do attempt to compare individual rat's sensory noise to their evidence discounting but find no compelling relationship. Instead, they argue that the average value of sensory noise in the population approximately matches the average value of discounting in the population. To waive away the meaningfulness of the lack of a relationship, the authors point out that the observed variability in evidence discounting across rats has negligible effect on overall task accuracy.

We agree with the reviewer that the best test of our claims would be to systematically vary the sensory noise levels. Unfortunately modifying the auditory cues and testing the level of sensory noise is outside the scope of this study.

We modified the text of the discussion to address this concern, adding:

“Our study has several potential limitations. First, the presence of sensory noise complicates the analysis of behavioral and neural data. A more direct test of our claim that sensory noise explains rats’ integration timescales would be an experimental modulation of sensory noise by modifying the acoustics of the stimuli or behavioral rigs. While such a manipulation would be insightful, it falls outside the scope of this study. Lowering the sensory noise would facilitate investigations with harder click rates, and potentially distinguish the nonlinear and linear discounting models. Second, this study tested a limited range of click rates and environmental volatilities, primarily focusing on one set of parameters. Rodent behavior may deviate from the optimal timescale in different parameter regimes where other factors influence behavior. Future work should investigate how evidence integration timescales change over a broader parameter regime.”

The authors forward their task design as a good basis from which to study the neurophysiology of evidence accumulation in a dynamic environment. The authors astutely point out the great opportunity for understanding neural mechanisms for learning and representing the decision-relevant statistics of the external world. Unfortunately, this task may not be well-suited to this goal. The level of sensory noise in this task is quite high (limiting localization accuracy to only 65%), meaning that much of animal’s representations and behavior is not predictable based on controlled external events. A task with much lower sensory noise would form a better basis for the desired neurophysiological investigations and would provide a good comparison point for the role of sensory noise.

While we share the reviewer’s concern that sensory noise complicates the analysis of neural data, we believe our task is still a suitable basis for analysis of neural coding and representations. Previous studies in our lab (Hanks et al 2015, *Nature*; Yartsev et al 2018, *under review*), utilized the same auditory clicks on the static version of this task to study neural coding in several brain structures. We therefore feel confident that our task is well suited for neural investigations.

We modified the text of the discussion to address this concern, adding:

“Our study has several potential limitations. First, the presence of sensory noise complicates the analysis of behavioral and neural data. A more direct test of our claim that sensory noise explains rats’ integration timescales would be an experimental modulation of sensory noise by modifying the acoustics of the stimuli or behavioral rigs. While such a manipulation would be insightful, it falls outside the scope of this study. Lowering the sensory noise would facilitate investigations with harder click rates, and potentially distinguish the nonlinear and linear discounting models.”

Reviewer #3 (Remarks to the Author):

In this work, Piet, El Hady, and Brody investigate the ability of rats to optimally integrate point process data in the form of auditory clicks for purposes of deciding about the underlying state of a changing world. They show that when misidentification of clicks is accounted for (and estimated from a separate study), rats’ ability to discount past information is optimal.

In one sense, this is an unsurprising result. That animals, especially highly overtrained ones, can learn to perform such a task optimally should not shock most readers. Nonetheless, this study is important for at least two reasons: First, in many cases, animals **can** fail to integrate information optimally, and understanding the circumstances under which this happens (or the circumstances under which behavior can be explained by constraints like noise floors) is valuable. Second, the literature on decision-making in dynamic environments

has proven fruitful for our understanding of the brain's online information processing, and as the authors note, their analysis opens up new avenues of investigation for their and similar paradigms.

I enthusiastically recommend publication with only minor quibbles. I also want to congratulate the authors on what I found to be a very lucid presentation of such technical material.

We thank the reviewer for the positive evaluation of our work. We respond in detail to each minor comment below.

Minor comments:

I. 210: It would help to elaborate on this a little more. A sentence or two recapitulating what the figure shows would help drive the point home.

We thank the reviewer for pointing out the ambiguity here, we have modified the text to more clearly reflect what we computed, and what it means.

"The optimal inference equation attempts to predict the hidden state. As the hidden state dynamically transitions, we expect the inference process to track, albeit imperfectly, the dynamic transitions. From the perspective of a subject this dynamic tracking leads to changes of mind in the upcoming choice. Through the optimal inference process we can predict the timing of changes of mind by looking for times when the sign of the inference process ($\text{sign}(a)$). We simulated the optimal inference agent on a large dataset of trials assuming either no sensory noise (black), or average rat sensory noise (pink). For both agents we computed the distribution of when changes of mind happen relative to changes in the hidden state of the trial. Figure 2D shows the predicted timing of changes of mind with and without sensory noise, and the temporal relationship to hidden state changes. As expected, hidden state changes trigger changes of mind with a temporal delay that increases with sensory noise. The presence of sensory noise slows the integration timescale, and thus slows the timing of changes of mind."

Eq 15: $\Delta L, t$ should be $\Delta_{\{L, t\}}$?

We thank the reviewer for pointing out this typo, it has been corrected.

Fig 5B: In the caption: isn't this pink, not orange?

We don't want to call it pink, because pink in previous figures is used to denote the optimal theory with sensory noise. We have instead referred to this panel as "light red."

Figure 6: It would be helpful if the supplement could show individuals.

Figure 29 in the supplement now shows each rat's discounting for blocks of trials as the environmental statistics were modified.

Figure 29. Discounting Rates change in response to environmental volatility. The discounting parameter for each rat in blocks of 7500 trials over changing environmental statistics. Each rat was moved from a hazard rate = 0.5Hz environment, to a static environment, and back to a 0.5 hz environment. Rat H034 was removed from training during the 0Hz block and thus did not perform the return to 0.5 hz trials. Rats H045 and H046 were first moved from a 1Hz environment to a 0.5 Hz environment before moving to a static environment.

Eqns 19-22: Eq 20 is just a further simplification of Eq 19, correct? Shouldn't these just be one equation with aligned right-hand side?

The reviewer is correct that Eq. 20 is a simplification of Eq. 19, and Eq. 22 is a simplification of Eq. 21. We modified the equation alignment and numbering to reflect this. Now the expression for the mean of the distribution is numbered Eq. 19, and the expression for the variance of the distribution is Eq. 20.

Eq 26: The lead-up to this is a bit terse. I managed to fill in the steps of the derivation, but it would help most readers if this was laid out a bit more explicitly. Some of this is spelled out in the supplement. A few sentences here and a pointer to the relevant sections of the supplement should do it.

We thank the reviewer for pointing out the ambiguity here, we added details to the supplementary materials and added a pointer in the main methods

Main methods:

Then, we computed what fraction of the probability mass would cross 0 to be registered as a click on the other side (See Supplemental Materials "Click mislocalization from model parameters").

We added this to the supplementary materials section on sensory noise

Click mislocalization from model parameters

Our analysis of sensory noise uses an estimation of the rate of click mislocalization. Both our estimate of mislocalization for our rats, and rats from Brunton 2013 et. al. are derived from parameters in the trial-by-trial model. Here we provide details on how we use those model parameters to estimate the click mislocalization, n .

The three model parameters that influence n are the sensory noise σ^2_s , the adaptation strength ϕ , and the adaptation time constant τ_ϕ . First, we compute the average level of sensory adaptation on each click, $\langle C \rangle$ by simulating the adaptation dynamics (Eq. 15) on a large dataset of trials. Given the average adaptation on each click, the sensory noise σ^2_s creates a Gaussian distribution of effective click amplitudes with mean $\langle C \rangle$, and variance $\sigma^2_s \langle C \rangle$ (Brunton, 2013). To compute what fraction of the probability mass would cross 0 to be registered as a click on the other side we use the Gaussian cumulative distribution function. For a standard Gaussian $N(\mu, \sigma^2)$:

....Eq 39 here...

Plugging in the adapted mean and variance for the average click, and $x = 0$:

....Eq 40 here...

Figure 20: It would be better if each plot either used the correct character (λ is rendered, but “ ϕ ” is not) or had a more informative name. I’m not sure what “ss” is.

We apologize the model parameters were not correctly rendered, we have corrected the figure (Now Figure 25).

REVIEWERS' COMMENTS:

Reviewer #1 (Remarks to the Author):

We thank the authors for an extremely thorough response to our (and all the referees') comments on the manuscript. I still have a lingering question concerning their approach, which I simply want the authors to consider (it does not require revision of the paper). This is a fine piece of work, and I am happy to support publication in Nature Communications at this point.

The authors may do well to examine just how strongly the normative model depends on trial length and time since the last change point assuming there is no sensory noise. For high enough SNR, there is a very rapid change in belief at changepoints. Thus, the assumption of sensory noise is crucial for the normative (and associated linear) model to give with experimental performance observed. I know you used Brunton et al (2013) to find 'n', but it is not necessarily true that the parameters of rats' performance on the static environment task will match with those on a dynamic environment task. The 'n' may be different in either case. If both λ and 'n' are then allowed to be free parameters, there is likely a manifold of those values that will allow models to match performance. I guess Fig. 4 breaks this degeneracy to some degree, but it seems like one could do a more thorough analysis (but this is outside the scope of the current paper).

Reviewer #2 (Remarks to the Author):

The authors carefully, thoroughly, and more than adequately responded to every concern raised, leaving no reasonable doubt to this exceptional work.

Reviewer #3 (Remarks to the Author):

I appreciate the authors' careful responses to all reviewers and once again enthusiastically recommend publication.

Formatting note: In the following detailed responses, the reviewers concerns are in black text, and our responses are in blue text.

REVIEWERS' COMMENTS:

Reviewer #1 (Remarks to the Author):

We thanks the authors for an extremely thorough response to our (and all the referees') comments on the manuscript. I still have a lingering question concerning their approach, which I simply want the authors to consider (it does not require revision of the paper). This is a fine piece of work, and I am happy to support publication in Nature Communications at this point.

We thank the reviewer for this positive evaluation of our work.

The authors may do well to examine just how strongly the normative model depends on trial length and time since the last change point assuming there is no sensory noise. For high enough SNR, there is a very rapid change in belief at changepoints. Thus, the assumption of sensory noise is crucial for the normative (and associated linear) model to jive with experimental performance observed. I know you used Brunton et al (2013) to find 'n', but it is not necessarily true that the parameters of rats' performance on the static environment task will match with those on a dynamic environment task. The 'n' may be different in either case. If both λ and 'n' are then allowed to be free parameters, there is likely a manifold of those values that will allow models to match performance. I guess Fig. 4 breaks this degeneracy to some degree, but it seems like one could do a more thorough analysis (but this is outside the scope of the current paper).

The reviewer highlights an important point that the role of sensory noise is required for the normative model to match the rat behavior, and raises a concern that 'n' and λ might trade-off with each other. Our accumulation model (Fig 5) allows both λ and 'n' (albeit parameterized as σ^2_s , τ and ϕ) to be fit independently for each rat. The resulting model fits show λ and 'n' are consistent with the reverse correlation analysis, 'n' is consistent with the Brunton et al (2013) estimates, and critically, the λ parameter does not trade-off with other parameters, including 'n' (Supplementary Figure 22). Taken together, we feel confident in our conclusion that sensory noise determines the timescale for discounting. Future work will consider more detailed models of sensory noise.

Since the figure panel showing that λ does not trade off with other parameters is already in the manuscript and addresses the reviewer's concern, we have not further modified the manuscript regarding this issue (consistent with the reviewer's suggestion that their comment did not require revision of the paper).

Reviewer #2 (Remarks to the Author):

The authors carefully, thoroughly, and more than adequately responded to every concern raised, leaving no reasonable doubt to this exceptional work.

We thank the reviewer for this positive evaluation of our work.

Reviewer #3 (Remarks to the Author):

I appreciate the authors' careful responses to all reviewers and once again enthusiastically recommend publication.

We thank the reviewer for this positive evaluation of our work.